# Older adults are relatively more susceptible to impulsive social influence than young adults
Zhilin Su [1] ✉, Mona M. Garvert [2], Lei Zhang [1,3,4], Sanjay G. Manohar[5,6], Todd A. Vogel [1,3,4], Louisa Thomas[7], Joshua H. Balsters[7], Masud Husain [5,6,8], Matthew A. J. Apps [1,3,4,5,6,8] & Patricia L. Lockwood [1,3,4,5,6,8] ✉

People differ in their levels of impulsivity and patience, and these preferences are heavily influenced by others. Previous research suggests that susceptibility to social influence may vary with age, but the mechanisms and whether people are more influenced by patience or impulsivity remain unknown. Here, using a delegated inter-temporal choice task and Bayesian computational models, we tested susceptibility to social influence in young (aged 18–36, $N = 76$) and older (aged 60–80, $N = 78$) adults. Participants completed a temporal discounting task and then learnt the preferences of two other people (one more impulsive and one more patient) before making their choices again. We used the signed Kullback-Leibler divergence to quantify the magnitude and direction of social influence. We found that, compared to young adults, older adults were relatively more susceptible to impulsive social influence. Factor analyses showed that older adults with higher self-reported levels of affective empathy and emotional motivation were particularly susceptible to impulsive influence. Importantly, older and young adults showed similar learning accuracy about others' preferences, and their baseline impulsivity did not differ. Together, these findings suggest highly affectively empathetic and emotionally motivated older adults may be at higher risk for impulsive decisions, due to their susceptibility to social influence.

Humans vastly differ in how impulsive or patient they are (i.e., their willingness to wait for larger rewards in the future). These differences have profound economic, societal and psychiatric implications[1–4]. However, how impulsive or patient a person is can also be strongly influenced by the behaviours of those around them[5]. People often change their behaviours to emulate others, henceforth referred to as 'social influence'[5–8]. Understanding why and how people are susceptible to social influence, as well as identifying the nature of influence, is crucial at the individual and societal level, such as for political decision-making and social cohesion[9–11]. Social influence can also play a critical role in impulsivity[12–16]. Yet whether such susceptibility drives people to be more impulsive or more patient remains poorly understood.

Intriguingly, research suggests that susceptibility to social influence might differ across the lifespan. Adolescence, the period between the onset of puberty and the attainment of independence, is often associated with increased risk-taking, deeper need for social connection, and greater susceptibility to peer pressure[17]. Compared to young adults, adolescents have been shown to be more sensitive to peer influence and more likely to engage in risky behaviours when in the presence of others[18,19]. For example, a longitudinal study reported that susceptibility to social influence decreased across adolescence[16]. This reinforces the idea that people's inclination to be influenced by others may vary across different stages of life.

However, little is known about how ageing affects susceptibility to social influence. Understanding how susceptibility to social influence evolves in the latter part of life has significant implications for public policy, such as addressing the rising prevalence of misinformation amongst older adults[20]. Previous research suggests alternative hypotheses for how ageing is

[1]Centre for Human Brain Health, School of Psychology, University of Birmingham, Birmingham, B15 2TT, UK. [2]Faculty of Human Sciences, Junior professorship of Neuroscience, University of Würzburg, 97070 Würzburg, Germany. [3]Institute for Mental Health, School of Psychology, University of Birmingham, Birmingham, B15 2TT, UK. [4]Centre for Developmental Sciences, School of Psychology, University of Birmingham, Birmingham, B15 2TT, UK. [5]Nuffield Department of Clinical Neurosciences, University of Oxford, Oxford, OX3 9DU, UK. [6]Department of Experimental Psychology, University of Oxford, Oxford, OX2 6GG, UK. [7]Department of Psychology, Royal Holloway, University of London, Surrey, TW20 0EX, UK. [8]Wellcome Centre for Integrative Neuroimaging, University of Oxford, Oxford, OX3 9DU, UK. ✉e-mail: zhilinsu1312@gmail.com; z.su.1@pgr.bham.ac.uk; p.l.lockwood@bham.ac.uk

associated with such vulnerability. One possibility, according to the socio-emotional selectivity theory[21], is that socioemotional goals become more prominent in people's lives as they age. Therefore, older adults may demonstrate a heightened susceptibility to social influence compared to young adults. An alternative hypothesis is that older adults, drawing from their extensive life experiences and enhanced skills in reasoning about social conflicts[22], may have a greater capacity to resist social influence than their younger counterparts. Finally, to be influenced by others, we must be able to learn what others' preferences are. Older adults have been shown to have reduced reinforcement learning abilities when outcomes affect themselves[23]. However, when outcomes relate to other people, their learning is preserved[24]. This suggests that older adults could be equally susceptible to social influence as young people as they are able to accurately learn from social information.

A final aspect of the puzzle is that younger and older adults may already differ in their preferences for patience and impulsivity before any social influence has occurred. The nature of these differences is somewhat controversial. Some theories suggest that older adults are more impulsive than their younger counterparts[21], whereas others state that older adults appear more patient[25]. Empirically, studies have found evidence both for[26–29] and against[30–32] such differences. Yet a recent meta-analysis of 37 cross-sectional studies suggested no robust effect of ageing on temporal impulsivity[33], and others have indicated non-linear age effects[34]. However, individual studies do find differences between some group samples. Part of these differences between studies could stem from variations in susceptibility to social influence in the samples that they test.

To address these alternative hypotheses, we employed Bayesian computational models[35] to study the effect of ageing on susceptibility to impulsive and patient social influence, using a well-characterised task assessing intertemporal preferences. Two groups of participants (young adults aged 18–36 and older adults aged 60–80), completed a temporal discounting task (i.e., participants choosing between smaller-and-sooner rewards and larger-and-later rewards according to their preferences) and then learnt about the preferences of two other people, one who was more impulsive, and the other who was more patient, before making their own discounting choices again (cf [14,15]). Participants also completed neuropsychological tests and self-report measures of socio-affective traits to account for potential individual differences in social conformity.

## Methods
### Participants
We recruited 80 young participants (aged 18–36) and 81 older participants (aged 60–80) to take part in this study. Participants were recruited from university databases, social media, and the community for both age groups to make sure participants were matched as closely as possible. Our exclusion criteria included current or previous study of psychology. Additionally, all individuals were without a history of neurological or psychiatric disorder, had normal or corrected-to-normal vision, and specifically for the older participants, scored above the threshold on the Addenbrooke's Cognitive Examination (with a cut-off score of 82), indicating no potential risk for dementia[36]. This sample size gave us 87% power to detect a significant interaction effect between age group and other's preference, as determined through a simulation-based power analysis[37].

Four young and three older participants were excluded from all analyses due to: diagnosis of a neuropsychiatric disorder at the time of testing (one young participant); previous study of psychology (two young participants); potential risk for dementia (one older participant); and failure to complete the task (one young and two older participants). This left a final sample of 154 participants, 76 young participants (31 men & 45 women aged 18–36, mean = 23.1) and 78 older participants (37 men & 41 women aged 60–80, mean = 70.0). We did not collect data on race or ethnicity. One participant from each age group was missing data on the self-report questionnaire measures and were excluded from the relevant analyses. In the final sample, eight young and four older participants had two agents with

similar patient preferences. Data from these participants was excluded from all analyses involving the agent with impulsive preferences, as there was no available data. Similarly, four young and ten older participants in the final sample had two agents with similar impulsive preferences. Their data was also excluded from analyses involving the agent with patient preferences due to a lack of data.

Participants were paid at a rate of £10 per hour and were told they would receive an additional bonus based on a randomly chosen trial from the experiment: the bonus amount would be rewarded after the specified delay, unless immediately. Actually, participants were paid a randomly selected bonus ranging from £1 to £10 on the day of testing and were informed that a trial had been selected. All participants provided written informed consent, and ethical approval of this study was granted by the University of Oxford Medical Sciences Interdivisional Research Ethics Committee. The study was not preregistered.

### Delegated inter-temporal choice task
Participants completed a delegated inter-temporal choice task where they learnt about impulsive and patient others after completing their own temporal discounting preferences (Fig. 1A). In this task, participants made a series of decisions between two offers. One offer was a smaller amount of money paid immediately (*today*), and the other offer was a larger amount of money paid after a variable delay period. The amount varied between £1 and £20, and the delay period ranged from 1 to 90 days (this was dynamically adjusted in the *Self* blocks). The two offers were presented at the same time, and the position of the immediate offer and delayed offer on the screen was randomised on a trial-by-trial basis. The experiment was subdivided into five blocks of 50 trials (*Self1, Other1, Self2, Other2, Self3*), with a self-paced break after 25 trials in each block, resulting in 250 trials overall (Fig. 1A). Participants were informed that the decisions they would see were those of previous participants who had already taken part in the study. In fact, these choices were computer generated as described below. No participant reported to the experimenter that they disbelieved the choices they observed were from other people. We further probed whether they had any disbelief in a post-study survey by asking if they had any questions or concerns about the task they completed. Both checks further demonstrated the validity of our task.

On trials in the *Self* blocks, (i.e., the first, third, and fifth blocks), participants were instructed to choose the preferred offer according to their true personal preferences, as they believed that one of these decisions would be honoured as their bonus payment. On trials in the *Other* blocks (i.e., the second and fourth blocks), participants were instructed to make decisions on behalf of the two other people, with the understanding that these choices were previously made by two other participants. The behaviours of these two people were simulated based on the participant's own choices in the *Self1* block. Participants received feedback on their choices, enabling them to learn the intertemporal preferences of the other agents (see below *Simulation of the other agents' choices*). The correct choices were defined as those with higher values estimated from the hyperbolic model, given a discount rate. Two gender-matched names (or two randomly chosen names for participants who did not specify their gender) were selected to represent these two other people. The participants were informed that their choices for the others were not communicated to the other people and did not have any consequences for either themselves or the other people. The task was presented in MATLAB 2012a (The MathWorks Inc) using the Cogent 2000 v125 graphic toolbox (software developed by the University College London; used to be available at www.vislab.ucl.ac.uk/Cogent/).

### Computational modelling
Participants' choices were used to estimate their discount rates separately for each experimental block using a standard hyperbolic discounting model[38]:

$$V_{LL} = \frac{M_{LL}}{1 + KD} \tag{1}$$

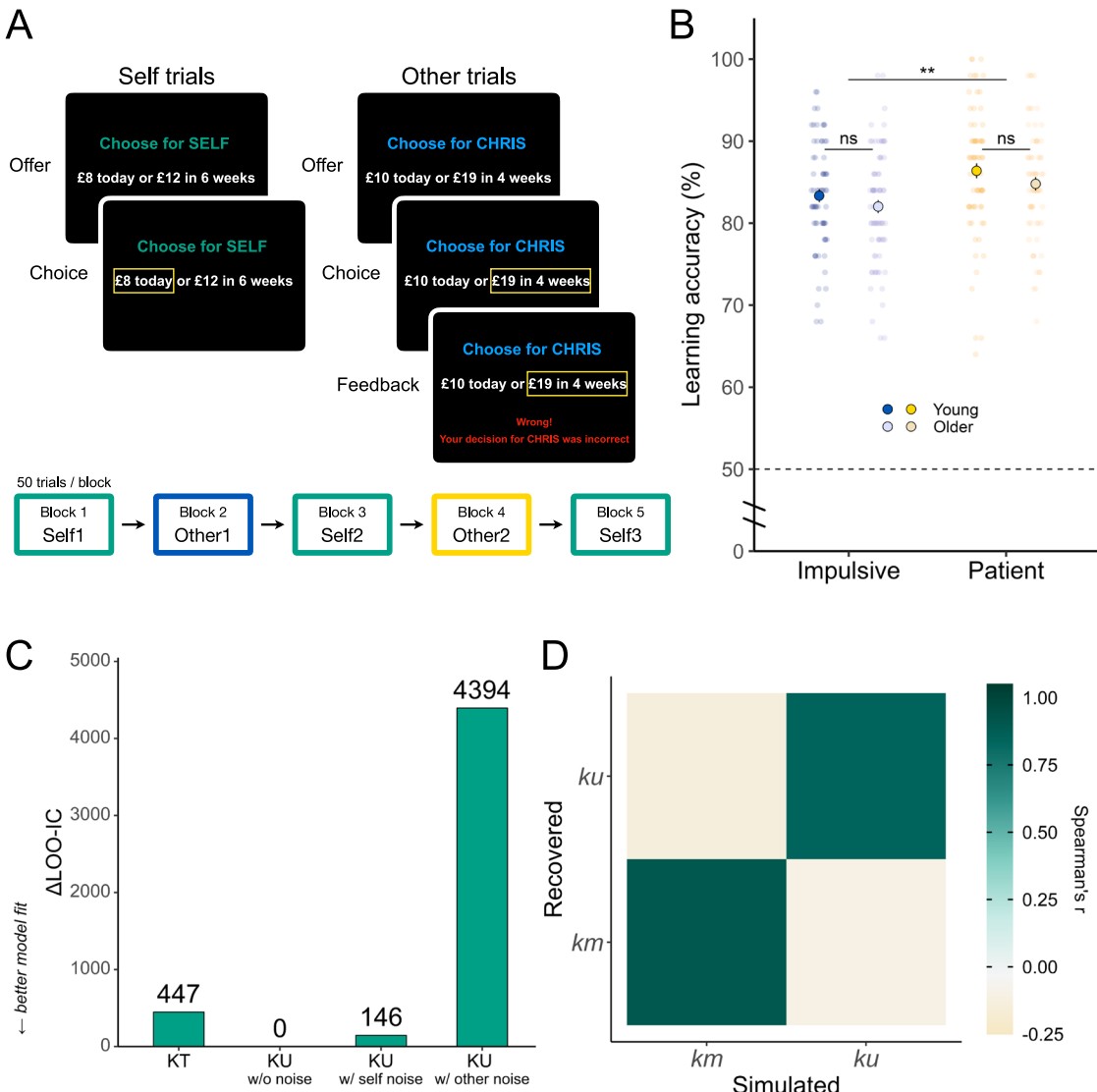

**Fig. 1 | Delegated inter-temporal choice task, learning performance, and model diagnostics. A** The trial structure in *Self* and *Other* blocks. On *Self* trials, participants were instructed to choose their preferred option between one offer which had a smaller amount of money paid immediately (smaller-and-sooner offer, *SS*) and the other offer which had a larger amount of money paid after a variable delay period (larger-and-later offer, *LL*). They were incentivised to indicate their true preferences by being informed that one of these decisions would be honoured as their bonus payment. On *Other* trials, participants were instructed to learn the preferences of the other two people, with the understanding that these choices were previously made by two other participants. Participants received feedback on their choices, enabling them to learn the intertemporal preferences of the other agents. The experiment was subdivided into five blocks of 50 trials (*Self1*, *Other1*, Self2, *Other2*, *Self3*), with a self-paced break after 25 trials in each block, resulting in 250 trials overall. The order of the other agents' preferences (*more impulsive* vs *more patient*) was counterbalanced across participants. **B** Comparison of learning accuracy shows that an equivalent learning performance of the other agents' preferences between the two age groups (no main effect of age group: $b = -0.01$, 95% CI = [−0.04 0.01], $Z = -1.22$, $P = 0.22$, $BF_{01} = 1.56$). Additionally, both young and older adults exhibited better learning of

the patient agents' preferences (significant main effect of other's preference: $b = 0.03$, 95% CI = [0.008 0.05], $Z = 2.71$, $P = 0.007$). Sample sizes differ across conditions due to the unavailability of relevant data for some participants ($N = 68$ for young impulsive, $N = 72$ for young patient, $N = 74$ for older impulsive, and $N = 68$ for older patient). Big circles with bordered lines represent the mean, and error bars are the standard error of the mean, dots are raw data, and the asterisks represent the significant main effect of other's preference from the linear mixed-effects model. Note that the axis includes a discontinuity between 0% and 50% to highlight the range of observed data more clearly. The dashed line at 50% indicates the chance level of performance. **\*\***$P < 0.01$; ns: not significant. **C** ΔLOO-IC (leave-one-out information criterion) relative to the winning model (KU model without noise parameters). **D** Parameter recovery. The confusion matrix represents Spearman's Rho correlations between simulated and recovered (fitted) parameters. Both *km* and *ku* exhibited strong positive correlations between their true and fitted values, with all $r_s > 0.85$. The entire parameter recovery procedure was iterated 20 times, with the Spearman's Rho correlation coefficients being averaged using Fisher's *Z*-transformation.

where $V_{LL}$ is the subjective value of a larger-and-later offer, $M_{LL}$ is the objective magnitude of the offer, $D$ is the delay period, and $K$ is a participant-specific hyperbolic discount rate that quantifies the devaluation of larger-and-later offers by time. The subjective value of a smaller-and-sooner offer ($V_{SS}$) will always correspond to its objective magnitude ($M_{SS}$) since the delay period is 0. Previous studies[1] have shown that the population tend to have an approximately normal distribution of $k = \log_{10}(K)$. Therefore, all reported

analyses are based on $k$, the log-transformed measure of $K$. When $k \to -\infty$, individuals tend not to discount delayed offers, evaluating an option solely based on its objective magnitude. As $k \to 0$, individuals become increasingly sensitive to delay periods and discount delayed offers more steeply. We considered a set of models applicable to our paradigm based on previous work[14] examining social influence to constrain our model space and for the results to be comparable across studies.

**Preference-temperature (KT) model.** During the experiment, the preference-temperature (KT) model was used to approximate participants' behaviours in the *Self1* block and simulate the choices of other agents. The KT model supposes that each participant possesses a distinct true discount rate. Within this model, the following softmax function was used to convert the difference in subjective values between the two offers ($V_{LL} - V_{SS}$) on each trial into choice probability for choosing the delayed offer:

$$P_{LL} = \frac{1}{1 + e^{-T(V_{LL} - V_{SS})}} \tag{2}$$

where $T$ is a participant-specific inverse temperature parameter that characterises the noisiness of an individual's decisions. A lower value for $T$ results in greater non-systematic variations around the indifferent point, which is the point at which both offers are equally preferred. In the *Self1* block during the experiment, the free parameter $k$ values were set between $-4$ and $0$, and the $\log_{10}(T)$ parameter (represented as $t$) values were set within the range of $-1$ and $1$.

**Preference-uncertainty (KU) model.** Contrary to the previously mentioned KT model, the preference-uncertainty (KU) model follows Bayesian inference, positing that participants' discount rate should be considered as a normal probability distribution rather than a single true value[14]. On each trial, participants sample a value of $k$ from a participant-specific normally-distributed discounting distribution that was updated on a trial-by-trial basis:

$$P_k = \mathcal{N}(k; km, ku^2) \tag{3}$$

where free parameters $km$ and $ku$ represent the mean and standard deviation of the normal distribution, respectively. The parameter $km$ indicates temporal impulsivity (i.e., how impulsive or patient a person is), while $ku$ indicates preference uncertainty (i.e., how certain a person is about their one preference). Participants will choose the offer whose subjective value is higher in a deterministic way. Derived from the Eq. (1), participants will choose the delayed offer if and only if $k < \log_{10}[(M_{LL} / M_{SS} - 1) / D]$; the choice probability for choosing the delayed offer given a single sample value from the discounting distribution of Eq. (3) is:

$$P_{LL} = \Psi\left(\log_{10}\left[(M_{LL}/M_{SS} - 1)/D\right]; km, ku^2\right) \tag{4}$$

where $\Psi$ denotes the cumulative distribution function of the normal distribution.

**Simulation of the other agents' choices**
The behaviours of the two other agents were simulated using the participants' baseline discount rates, which were estimated with the preference-temperature (KT) model in the first experimental block. More specifically, the other agent's choices were generated by a simulated hyperbolic discounter whose discount rate $k$ was either plus one (*more impulsive*) or minus one (*more patient*) from the participant's own baseline $k$ in the *Self1* block. Crucially, the choices of the simulated hyperbolic discounter were slightly noisy, as the subjective value of offers was translated to a choice probability using a softmax function (with the inverse temperature parameter $t = 1$). The order of the other agents' preferences (*more impulsive* vs *more patient*) was counterbalanced across participants.

**Signed Kullback-Leibler divergence**
The Kullback-Leibler divergence ($D_{KL}$), a measure of the discrepancy between two probability distributions[39], was used to quantify the change in participants' discount rates ($k$) after learning about the other agents. $D_{KL}$ is

defined as follows:

$$D_{KL}(P||Q) = \int_{-\infty}^{\infty} p(x)\log_{10}\left(\frac{p(x)}{q(x)}\right)dx \tag{5}$$

where $P$ and $Q$ are distributions of a continuous random variable defined on a sample space ($\mathcal{X}$) and $p$ and $q$ denote the probability densities of $P$ and $Q$. In this study, we used $D_{KL}$ to quantify the divergence in the posterior distributions of $k$ at the end of two consecutive *Self* blocks. $D_{KL}$ was signed for the further analyses[15]. Positive signed $D_{KL}$ values signify a shift in participants' discounting preferences towards those of the other agents, while negative signed $D_{KL}$ values indicate a shift away from them, compared to the baseline discounting preferences (see Fig. 2C):

$$\text{Signed } D_{KL} = \begin{cases} D_{KL}, & if \; \dfrac{km_{\text{other},i} - km_{\text{self},1}}{km_{\text{self},i+1} - km_{\text{self},1}} > 0 \\[2ex] -D_{KL}, & if \; \dfrac{km_{\text{other},i} - km_{\text{self},1}}{km_{\text{self},i+1} - km_{\text{self},1}} < 0 \end{cases} \tag{6}$$

where $km$ represents the mean of discounting distribution estimated using the KU model, and the subscript $i$ denotes the number of *Other* blocks (i.e., 2 or 4). For example, if a participant's discounting preference shifts to be more negative (i.e., more patient) after exposure to the discounting preference of a patient other agent, this would be reflected by a positive signed $D_{KL}$ value. More specifically, if the difference between the other and self baseline $km$ is negative, and the difference between self after observation and self baseline $km$ is also negative (i.e., when the differences are of the same sign), then the signed $D_{KL}$ is positive, which means that the participant becomes more similar to others. Conversely, negative signed $D_{KL}$ values signal a divergence in the participants' discounting preferences from those of the other agents. For example, if the difference between the other and self baseline $km$ is positive, while the difference between self after observation and self baseline $km$ is negative (i.e., when the differences are of opposite signs), then signed $D_{KL}$ is negative, which indicates that the participant becomes more dissimilar to others.

**Optimisation of choice pairs**
In order to ensure precise estimation of participants' discounting preferences, choice pairs for all *Self* trials were generated by alternating between two approaches: generative and adaptive methods (in the framework of KT model). The generative method involved generating every possible combination of amounts and delays for the choice pairs. In each *Self* block, 25 trials (i.e., half of the trials in each *Self* block) were chosen to closely align with the indifference points of 25 hypothetical participants, with $k$ values evenly spread across the range of $-4$ to $0$[13,15,40]. It was an efficient but relatively imprecise way to estimate participants' discounting parameters. The remaining 25 trials in each *Self* block were generated using an adaptive method that leveraged a Bayesian framework to yield accurate estimations of the discounting parameters. Previous studies have demonstrated that this method is capable of generating more reliable estimates of the $k$ value while requiring fewer trials[41,42]. The individual's initial prior belief regarding $k$ was set as a normal distribution with a mean of $-2$ and a standard deviation of 1, while $t$ was set to 0.3. Following each decision made by the participant, their belief distribution about $k$ was updated using Bayes' theorem. Subsequently, choice pairs were generated to probe our estimate of participants' indifference point, which was based on the expected value of the current posterior distribution of $k$.

In every *Other* block and for the parameter recovery, all of the choice pairs were generated using the generative method. The options presented to participants were specifically designed to closely align with the indifference points of 50 hypothetical participants, with $k$ values evenly distributed across the range of $-4$ to $0$.

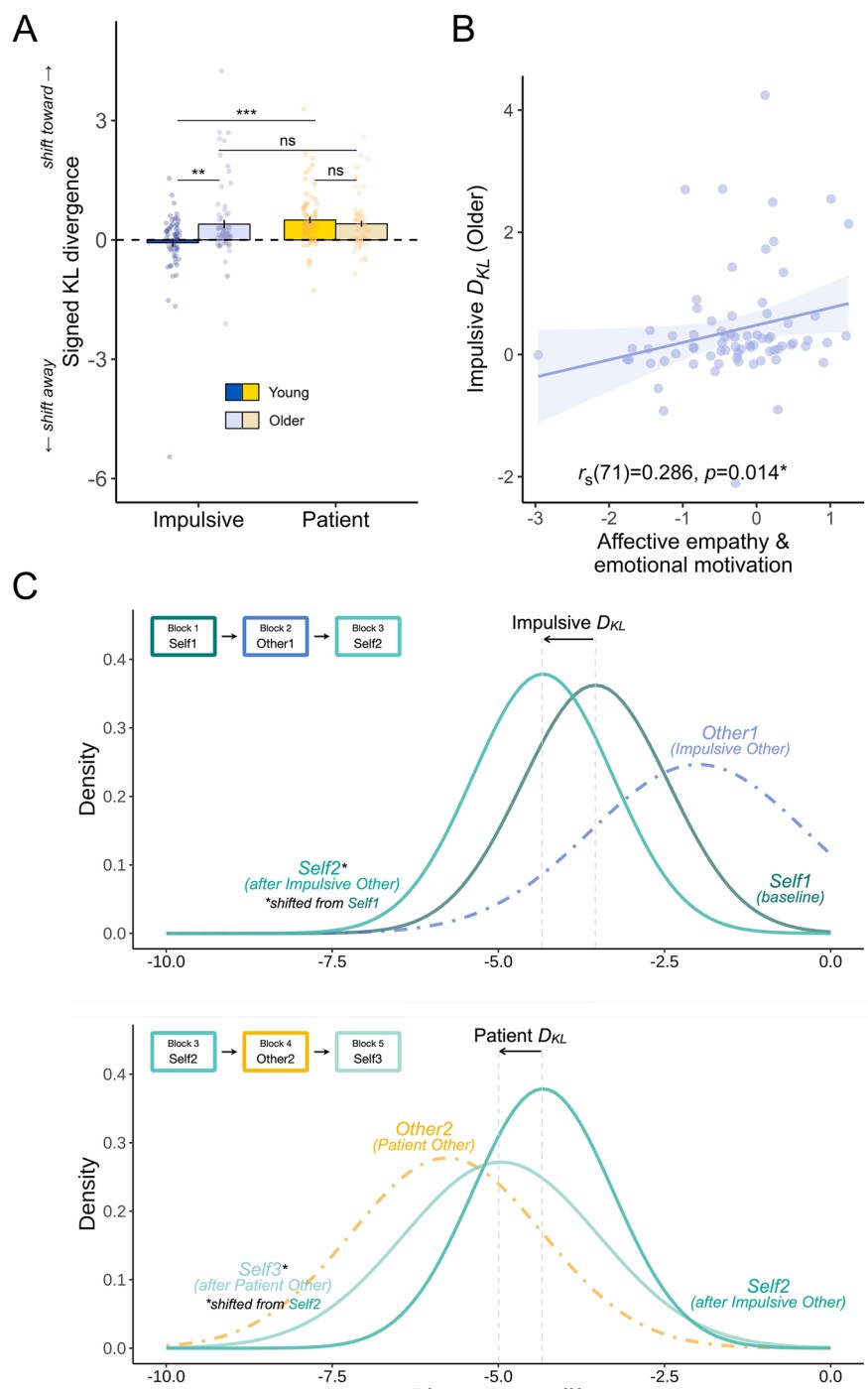

**Fig. 2 | Susceptibility to social influence quantified by the signed Kullback-Leibler divergence ($D_{KL}$). A** Older adults were more influenced by impulsive social influence than young adults ($W = 1861$, $Z = -2.67$, $r_{(140)} = 0.22$ [0.07 0.38], $P = 0.008$). In contrast, older and young adults demonstrated similar susceptibility to patient social influence ($W = 2723$, $Z = -1.15$, $r_{(138)} = 0.10$ [0.01 0.25], $P = 0.252$, $BF_{01} = 3.30$). Sample sizes differ across conditions due to the unavailability of relevant data for some participants ($N = 68$ for young impulsive, $N = 72$ for young patient, $N = 74$ for older impulsive, and $N = 68$ for older patient). Bars show group means, error bars are standard errors of the mean, dots are raw data, and asterisks represent significant two-sided between-group and within-group nonparametric $t$-tests. **$P < 0.01$; *** $P < 0.001$; ns: not significant. **B** A significant positive correlation was found between impulsive $D_{KL}$ and the factor 'Affective empathy & emotional motivation' scores amongst older adults ($r_{s(71)} = 0.29$ [0.06 0.48], $P = 0.014$). This positive correlation remained significant after correcting for multiple comparisons using the false discovery rate (FDR-corrected for three factor comparisons $P = 0.043$). $N = 73$ for this analysis as an additional participant's

self-report questionnaires were missing. **C** Example of shifts in self discounting distributions after learning about the preferences of an Impulsive Other and a Patient Other. (*upper*) In this example, the participant firstly completed a baseline block to assess their own baseline temporal preference (*Self1*, dark green solid line) before learning about the preference of an Impulsive Other (*Other1*, blue dashed line). After learning the preference of Impulsive Other, they completed another block making their own intertemporal choices (*Self2*, green solid line). For this participant, their preference shifted away from that of Impulsive Other ('*Impulsive $D_{KL}$*'), meaning that the participant's own temporal preference became less similar to that of the Impulsive Other (represented by a negative signed $D_{KL}$ value). (*lower*) Following this, the participant learnt about the preference of a Patient Other (*Other2*, yellow dashed line) before making their own intertemporal choice again (*Self3*, light green solid line). For this participant, their preference shifted towards that of Patient Other ('*Patient $D_{KL}$*'). The positive signed $D_{KL}$ value here means that the participant's preference became more similar to that of the Patient Other after observing their preference.

## Questionnaires

**Addenbrooke's Cognitive Examination (ACE-III).** The Addenbrooke's Cognitive Examination (ACE-III) was used to evaluate older adults for dementia[36]. The ACE assesses cognitive functioning across five domains: attention, memory, language, fluency, and visuospatial abilities. The ACE is scored on a scale of 0 to 100, and as a screening tool, a cut-off score of 82 out of 100 indicates significant cognitive impairment. All older participants included in the analyses scored above the cut-off score for dementia.

**Wechsler Test of Adult Reading (WTAR).** The Wechsler Test of Adult Reading (WTAR) was used to measure participants' general intelligence[43]. This test requires participants to pronounce 50 words that deviate from the typical grapheme-to-phoneme patterns. As such, the test evaluates reading recognition and prior knowledge of words, rather than the skill to use pronunciation rules. The WTAR scores show a strong correlation with the results from the Wechsler Memory Scale (WMS-III) and the Wechsler Adult Intelligence Scale (WAIS-III)[44]. The test is suitable for participants aged 16–89, covering our full sample.

**Autism Quotient (AQ).** The Autism Quotient was used to measure participants' traits associated with the autistic spectrum[45]. The AQ is scored using a Likert scale ranging from 1 (*definitely disagree*) to 4 (*definitely agree*)[46] instead of using the original binary scoring method, for better internal consistency and test-retest reliability. Five areas associated with the autistic spectrum are assessed: social skill, attention switching, attention to detail, communication, and imagination.

**Apathy Motivation Index (AMI).** The Apathy Motivation Index was used to measure participants' apathetic traits[47]. This scale consists of 18 items to measure three dimensions of individual differences in apathy-motivation: behavioural activation, social motivation, and emotional sensitivity. Participants were instructed to express their level of agreement with each item using a 5-point Likert scale ranging from 0 to 4. Every item is reversed scored, so higher values represent greater apathy.

**Toronto Alexithymia Scale (TAS).** The 20-item Toronto Alexithymia Scale (TAS) was used to measure participant's alexithymic traits, including difficulties in recognising, expressing, and describing one's emotions[48]. This scale includes three subscales: difficulty identifying feelings, difficulty describing feelings, and externally-oriented thinking. All items are rated on a Likert scale ranging from 1 (*strongly disagree*) to 5 (*strongly agree*).

**Self-Report Psychopathy scale (SRP).** The short form of the Self-Report Psychopathy scale (SRP-IV-SF) was used to measure participants' psychopathic traits[49]. This scale includes 29 items, with 7 items each assessing interpersonal, affective, lifestyle, and antisocial tendencies, plus an additional item, '*I have been convicted of a serious crime.*' Each item is scored on a five-point Likert scale ranging from 1 (*disagree strongly*) to 5 (*agree strongly*).

**Questionnaire of Cognitive and Affective Empathy (QCAE).** The Questionnaire of Cognitive and Affective Empathy (QCAE) was used to assess both the capacity to understand another person's emotions (*cognitive empathy*) and the ability to vicariously experience the affective experience of others (*affective empathy*)[50]. Items are rated on a 4-point Likert scale ranging from 1 (*strongly disagree*) to 4 (*strongly agree*).

**Delegated inter-temporal choice task-specific questionnaires.** Participants were asked four questions regarding their confidence in learning the other two agents' preferences, as well as their perceived similarity to these agents. Participants expressed their ratings by using a sliding scale that spanned from 0 (*not at all*) to 10 (*very confident/very similar*). All these self-report measures were collected through the Qualtrics platform (https://www.qualtrics.com/).

## Model fitting

We used R v4.2.1[51], Stan v2.32[52], and the RStan v2.21.7 package[53] for all model fitting and comparison. Stan employs Hamilton Monte Carlo (HMC), a highly efficient Markov Chain Monte Carlo (MCMC) sampling technique, to conduct full Bayesian inference and derive the true posterior distribution. Hierarchical Bayesian modelling was utilised to model participants' choices on a trial-by-trial basis. Hierarchical Bayesian modelling was adopted for its more stable and accurate parameter estimation[35]. In hierarchical Bayesian modelling, an individual-level parameter, denoted as $\phi$, was sampled from a group-level normal distribution, specifically:

$$\phi \sim \mathcal{N}\left(\mu_\phi, \sigma_\phi^2\right) \tag{7}$$

where $\mu_\phi$ and $\sigma_\phi$ are the group-level mean and standard deviation, respectively. The group-level parameters were specified with weakly-informative priors: $\mu_\phi$ conformed to a normal distribution centred around 0, with its standard deviation varied based on free parameters. Meanwhile, $\sigma_\phi$ adhered to a half-Cauchy distribution, having its location parameter set to 0, and its scale parameter varied according to free parameters. In the KT model, $k$ was set with a negative constraint, while $t$ was constrained to the range [-1 1]. In the KU model, $km$ had a negative constraint, whereas $ku$ had a positive constraint. Concerning the noise parameters, $\xi$ was restricted between [0 1], and $\tau$ fell within the range [0 10]. To ensure a more conservative estimation of all free parameters, the priors were reset at the beginning of each experimental block (i.e., the uninformative priors were used). We applied the hierarchical Bayesian modelling separately for young and older participants.

All group- and individual-level free parameters were simultaneously estimated through Bayes' theorem by integrating behavioural data. This approach allows for the consideration of both individual variability and overall group trends, leading to more robust and reliable parameter estimates[35]. We fitted each candidate model with four independent HMC chains. Each chain consisted of 2,000 iterations after an initial 2,000 warm-up iterations for the algorithm, resulting in 8,000 valid posterior samples. The convergence of HMC chains was evaluated through visual inspection (using the trace plot) and through the Gelman-Rubin $\hat{R}$ statistics[54]. For all free parameters in the winning model, $\hat{R}$ values were found to be close to 1.0, indicating satisfactory convergence.

## Model comparison and parameter recovery

For model comparison, we calculated the Leave-One-Out information criterion (LOO-IC) score for each candidate model[55], using the {*loo*} v2.5.1 package[56]. The LOO-IC score leverages the entire posterior distribution to provide a point-wise estimate for out-of-sample predictive accuracy in a wholly Bayesian manner. This method is more reliable than information criteria that are solely based on point-estimates, such as the Akaike information criterion (AIC) and the Bayesian information criterion (BIC)[55,57]. A lower LOO-IC score signifies superior out-of-sample predictive accuracy and better fit for a given model. The model with the lowest LOO-IC score was chosen as the winning model. Our winning model was the KU model without any additional noise parameters.

After model fitting, we confirmed the identifiability of parameters through parameter recovery. Let $\phi$ represent a generic free parameter in the winning model. We randomly drew a set of group-level parameters from the same weakly-informative prior group-level distribution used in model fitting. Here, $\mu_\phi$ and $\sigma_\phi$ denote the group-level mean and standard deviation, respectively:

$$\mu_\phi \sim \mathcal{N}(0, 3)$$
$$\sigma_\phi \sim \mathcal{HC}(0, 2) \tag{8}$$

where $\mathcal{HC}$ corresponds to the half-Cauchy distribution. Subsequently, we simulated 160 synthetic participants, deriving their parameters from this set of group-level parameters. For these 160 synthetic participants, their individual-level parameters, $\phi_i$, were sampled from a normal distribution using the corresponding group-level parameters:

$$\phi_i \sim \mathcal{N}\left(\mu_\phi, \sigma_\phi^2\right). \tag{9}$$

Next, we used the winning model as a mechanism to generate simulated behavioural data for our delegated inter-temporal choice task. In particular, we simulated decisions across 50 trials for each synthetic participant, using the choice pairs generated from the generative method (see the *Optimisation of choice pairs*). Then, we fitted our winning model to the simulated data in the same way as we did for our real participant data. Namely, we fitted the KU model (without any noise parameters) to the simulated individual data using HMC via Stan. This yielded posterior distributions for free parameters at both the group and individual levels. Finally, we calculated Spearman's Rho correlations between the simulated and recovered parameters at the individual level. The entire parameter recovery procedure was iterated 20 times, with the Spearman's Rho correlation coefficients being averaged using Fisher's $Z$-transformation.

## Statistical analysis
We used R v4.2.1 along with RStudio[58] to analyse the effect of age group and other's preference on the fitted model parameters and behavioural data. Linear mixed-effects models (LMM; '*lmer*' function from the {*lme4*} v1.1-33 package)[59] were used to predict individuals' learning accuracy, signed KL divergence values ($D_{KL}$), and scores from task-specific questionnaires (see Supplementary Methods for formula forms of models). We utilised linear mixed-effects models given their capability to account for the within-subject nature of the other's preference manipulation and their independence from parametric assumptions. For analysing learning accuracy, signed $D_{KL}$, and scores from task-specific questionnaires, the linear mixed-effects models incorporated fixed effects of age group (*older* vs *young*), other's preference (*patient* vs *impulsive*), and their interaction, along with a random subject-level intercept. An additional analysis of signed $D_{KL}$ also included participants' baseline $km$ (continuous covariates, centred around the grand mean) and its interaction with age group and other's preference (including the three-way interaction) as fixed terms. In another analysis controlling for general IQ, standardised scores on the WTAR were also included as a fixed term (without interacting with other terms). To compare learning accuracy to the chance level, we used right-tailed binomial exact tests against 50% ('*binom.test*' function from the {*stats*} v4.2.1 package). For simple and post hoc comparisons, we used two-sided paired and independent nonparametric tests ('*wilcox_test*' function from the {*rstatix*} v0.7.1 package)[60] for outcome variables that did not adhere to the normality assumptions. The normality was formally tested using the Shapiro-Wilk normality test ('*shapiro_test*' function from the {*rstatix*} v0.7.1 package). Effect sizes and confidence intervals for such nonparametric tests were determined using the '*wilcox_effsize*' function (from the {*rstatix*} v0.7.1 package as well). Correlations of signed $D_{KL}$ with self-reported socio-affective traits were calculated with Spearman's Rho nonparametric tests ('*rcorr*' function from the {*Hmisc*} v4.7-2 package; '*corr.test*' function from the {*psych*} v2.4.3 package)[61,62]. Additionally, we conducted $Z$ tests to compare these independent correlations ('*cocor.indep.groups*' function from the {*cocor*} v1.1-4 package)[63], and applied false discovery rate (FDR) correction for multiple comparisons across these correlations ('*p.adjust*' function from the {*stats*} v4.2.1 package). To account for general IQ and executive functions (attention and memory) when assessing the relationship between older adults' impulsive signed $D_{KL}$ and self-reported socio-affective traits, we conducted partial correlations, each controlling for either standardised WTAR, ACE attention, or ACE memory scores. These partial correlations were determined using the correlations between residuals derived from linear regression analyses ('*corr.test*' function from the {*psych*} v2.4.3 package). To assess non-significant results, Bayes factors (BF$_{01}$) were computed using paired and independent nonparametric $t$-tests in JASP v0.17.3[64] with the default prior, using linear models with the JZS prior ('*lmBF*' function from the {*BayesFactor*} v0.9.12-4.4 package)[65], using nonparametric linear correlations with the help of data augmentation ('*spearmanGibbsSampler*' and '*computeBayesFactorOneZero*' functions fetched from the OSF: https://osf.io/gny35/)[66]. BF$_{01}$ for $Z$ tests following the Fisher's $Z$-transformation was computed using the '*BF*' function from the {*BFpack*} v1.2.3 package[67]. BF$_{01}$ quantifies the extent to which the data are more likely under the null hypothesis of no difference compared to the alternative hypothesis of a difference. Bayes factors were interpreted and reported using the language suggested by Jeffreys[68]. All figures of statistical analysis were produced using the {*ggplot2*} v3.4.2 package[69].

## Exploratory factor analysis
We performed an exploratory factor analysis on the questionnaire subscales using the '*fa*' function (from the {*psych*} v2.4.3 package) in R v4.2.1. We incorporated all the subscales from the Autism Quotient (AQ), Apathy-Motivation Index (AMI), Toronto Alexithymia Scale (TAS), Self-Report Psychopathy scale (SRP), and the Questionnaire of Cognitive and Affective Empathy (QCAE). To extract factor loadings, we used maximum likelihood estimation with an oblimin rotation. Regarding the determination of the number of factors (using the '*fa.parallel*' and '*vss*' functions from the {*psych*} v2.4.3 package), the Kaiser rule (eigenvalue > 1) pointed toward a 2-factor solution, the very simple structure (VSS) complexity 2 criterion implicated a 3-factor solution, examination of the scree plot indicated a 3-factor solution, and parallel analysis suggested a 4-factor solution. After weighing parsimony and interpretability of the latent structure, we settled on the 3-factor solution. This 3-factor latent structure explained 50.55% of the variance in the measures, with moderate correlations with each other (highest $r = 0.25$). Individual scores for each factor were calculated using Thurstone's at the participant level. These scores were subsequently correlated with the signed KL divergence using Spearman's Rho correlation coefficients.

## Reporting summary
Further information on research design is available in the Nature Portfolio Reporting Summary linked to this article.

## Results
We analysed the behaviour of 76 young (aged 18–36) and 78 older adults (aged 60–80) who completed a temporal discounting task (Fig. 1A), neuropsychological tests, and self-report measures of socio-affective traits (see Methods). In the task, participants completed a block to assess their own temporal discounting preferences and were then introduced to the preferences of two other players who ostensibly previously took part in the same temporal discounting task. One of these players was constructed to be more impulsive than the participant themselves, and one who was constructed to be more patient, compared to their own baseline preferences, and these 'others' were presented in a counterbalanced order (see Methods). No participant reported disbelief that the preferences that they learnt were not genuinely those from other people.

Groups were matched as closely as possible on neuropsychological testing, IQ and demographics. All older adults were free of dementia (assessed by the Addenbrooke's Cognitive Examination (ACE)[36]). The groups did not differ in terms of gender ($\chi^2(1) = 0.45$, $P = 0.50$), years of education ($W = 2602$, $Z = -1.10$, $r_{(150)} = 0.09$ [0.00, 0.26], $P = 0.27$, BF$_{01} = 5.06$), or standardised IQ test performance ($W = 2670$, $Z = -1.06$, $r_{(152)} = 0.09$ [0.00, 0.25], $P = 0.287$, BF$_{01} = 4.92$). IQ test performance was measured using age-standardised scores on the Wechsler Test of Adult Reading (WTAR)[43]. We conducted further control analyses, accounting for IQ test performance (using standardised WTAR scores, taken by both young and older adults), as well as memory and attention (based on the memory and attention subscales from the ACE, exclusive to older adults). These control analyses did not change our results, indicating that our findings were not attributed to IQ test performance or executive function (see Methods and Tables S1, S2).

## Older and young adults can both learn others' preferences accurately

To validate participants' ability to complete the task, we first examined whether they were able to learn the preferences of the other agents with different discounting preferences significantly above the chance (50%) (Fig. 1B). Both young and older adults exhibited learning performances above the chance level when learning about impulsive (right-tailed exact binomial test against 50%: young group mean = 83%, proportion = 1.00 [0.96, 1.00], $P < 0.001$; older group mean = 82%, proportion = 1.00 [0.96, 1.00], $P < 0.001$) and patient others (young group mean = 86%, proportion = 1.00 [0.96, 1.00], $P < 0.001$; older group mean = 85%, proportion = 1.00 [0.96, 1.00], $P < 0.001$), indicating all age groups were capable of learning in the task.

Next, we examined whether there were preference-specific differences in learning between the two age groups. Overall, participants were more accurate at learning the preferences of patient compared to impulsive others ($b = 0.03$, 95% CI = [0.01 0.05], $Z = 2.71$, $P = 0.007$), an effect that did not significantly differ by age group, with only anecdotal evidence supporting no difference (main effect $b = -0.01$, 95% CI = [−0.04, 0.01], $Z = -1.22$, $P = 0.22$, $BF_{01} = 1.56$; age group × other's preference interaction $b = -0.001$, 95% CI = [−0.03, 0.03], $Z = -0.08$, $P = 0.94$, $BF_{01} = 6.06$).

After the task, participants completed self-report measures probing their confidence in learning. Here we observed that older adults reported less confidence in their learning ability ($b = -0.59$, 95% CI = [−1.00, −0.18], $Z = -2.82$, $P = 0.005$), across both patient and impulsive others (main effect $b = 0.21$, 95% CI = [−0.14, 0.55], $Z = 1.17$, $P = 0.24$, $BF_{01} = 3.73$; interaction $b = -0.10$, 95% CI = [−0.58, 0.39], $Z = -0.38$, $P = 0.70$, $BF_{01} = 5.90$), despite similar learning accuracy performance. In summary, learning performances were comparable across both age groups, with older adults reporting less confidence in their learning ability.

## Baseline impulsivity does not differ with age

Next, we used computational models of hyperbolic discounting[38], a well-established framework to explain delay discounting behaviour, to estimate participants' baseline temporal discounting preferences. Models were fitted using hierarchical Bayesian modelling[70,71], compared using out-of-sample cross validation, and verified using parameter recovery. We tested different models that varied based on non-Bayesian (Preference-Temperature (KT)) and Bayesian (Preference- Uncertainty (KU)) temporal preferences and choice variability. While the KT model assumes participants' discount preference to be a single value, the KU model computes discount preferences as a distribution. Based on recent studies examining these different formulations of discounting[14], we evaluated four candidate models (see Methods for full details):

(i) Preference-temperature (KT) model: a single discount rate ($k$) and an inverse temperature parameter ($t$) for the softmax function.
(ii) Preference-uncertainty (KU) model: a mean ($km$) and a standard deviation ($ku$) of the discounting distribution.
(iii) KU model with self-noise parameter: $km$, $ku$, and with a self-noise parameter ($\xi$):

$$P'_{LL,\text{self}} = P_{LL,\text{self}}(1 - \xi) + \xi/2 \tag{10}$$

(iv) KU model with other-noise parameter: $km$, $ku$, and with an other-noise parameter ($\tau$) to account for the choice stochasticity:

$$P'_{LL,\text{other}} = \frac{P_{LL,\text{other}}^{\frac{1}{\tau}}}{P_{LL,\text{other}}^{\frac{1}{\tau}} + \left(1 - P_{LL,\text{other}}\right)^{\frac{1}{\tau}}} \tag{11}$$

We found that participants' choices were best characterised by the KU model without any additional noise parameters (i.e., model ii). This model had the lowest LOO-IC score (leave-one-out information criterion, Fig. 1C), with parameters from the winning model showing excellent recovery (all

$r_s > 0.85$; Fig. 1D). Furthermore, the posterior predictive prediction also accurately replicated the key patterns in our behavioural data (see Supplementary Methods and Fig. S1). Additionally, the parameters estimated from the winning model were highly correlated with those used to generate simulate choices (see Methods and Fig. S2). All of these verified the validity of our winning model. These parameters $km$ and $ku$ serve as crucial indicators of temporal impulsivity and preference uncertainty, respectively. We therefore used this winning model to estimate participants' baseline discounting preference prior to learning. We found no credible evidence of difference in either mean (young group mean [SE] = −4.79 [0.22], older group mean [SE] = −5.16 [0.25]; independent Wilcoxon signed-rank test; $W = 3243$, $Z = -1.01$, $r_{(152)} = 0.08$ [0.005 0.23], $P = 0.314$, $BF_{01} = 3.47$; Fig. S3) or standard deviation (young group mean [SE] = 1.37 [0.06], older group mean [SE] = 1.47 [0.06]; $W = 2481$, $Z = -1.74$, $r_{(152)} = 0.14$ [0.01, 0.29], $P = 0.081$, $BF_{01} = 2.31$) of the discounting distribution between age groups (see Table S3 for results of all the experimental blocks). In addition, Bayes factors indicated strong evidence of no difference in the mean between the two age groups ($BF_{01} = 3.47$), whereas there was only anecdotal evidence supporting the null for the standard deviation ($BF_{01} = 2.31$). This shows that there was no credible evidence of difference in baseline impulsivity between the two age groups.

## Older adults are relatively more susceptible to impulsive social influence than young adults

After validating there was no credible evidence of difference in baseline temporal preferences between young and older adults, we subsequently examined their susceptibility to social influence using signed KL divergence ($D_{KL}$)[15,39] (see Methods). $D_{KL}$ quantifies the discrepancy between two probability distributions. This metric compares the entire probability distributions, rather than just summary statistics or point estimates from those distributions. In our analysis, $D_{KL}$ was signed to reflect the direction of shifting in the discounting distributions compared to the baseline (see Methods and Fig. 2C). Positive signed $D_{KL}$ values indicate a shift towards other people's discounting preferences (i.e., become more similar to others), while negative values suggest a shift away from them compared to baseline preferences.

We tested whether there were group differences in susceptibility to social influence when learning about impulsive and patient others. A linear mixed-effects model of signed $D_{KL}$ revealed that there was a significant interaction between age group and other's preference ($b = -0.56$, 95% CI = [−0.93, −0.20], $Z = -3.03$, $P = 0.002$, Fig. 2A). Strikingly, older adults were more influenced by impulsive social influence than young adults ($W = 1861$, $Z = -2.67$, $r_{(140)} = 0.22$ [0.06, 0.38], $P = 0.008$). In contrast, older and young adults demonstrated similar susceptibility to patient social influence ($W = 2723$, $Z = -1.15$, $r_{(138)} = 0.10$ [0.01, 0.25], $P = 0.252$, $BF_{01} = 3.30$).

While older adults learnt about the patient others better, they remained equally susceptible to the influence of both impulsive and patient others (paired Wilcoxon signed-rank test; $V = 886$, $Z = -1.03$, $r_{(62)} = 0.13$ [0.01, 0.38], $P = 0.305$, $BF_{01} = 5.49$). This finding was supported by strong evidence of no difference ($BF_{01} = 5.49$). In contrast, young adults were more influenced by patient than impulsive others ($V = 469$, $Z = -3.82$, $r_{(62)} = 0.48$ [0.27 0.66], $P < 0.001$), and they also learnt better about patient others. There was no significant correlation between participants ability to learn the preference of the other people and how much they shifted towards them (all $|r_s| s < 0.14$ and all $Ps > 0.27$, Table S4), suggesting group differences between young and older adults were not driven by possible individual differences in learning ability. Additionally, we replicated this behavioural pattern using the model-free index (see Supplementary Methods, Fig. S4, and Table S5).

As an additional control analysis, we also examined whether people's vulnerability to social influence depends on their baseline impulsivity. Although we observed no between-group difference in baseline discounting, we wanted to ensure the stronger susceptibility to impulsive others amongst older adults was not driven by individual differences in the baseline impulsivity. A linear mixed-effects model showed no significant interactions

between baseline discounting and any of our effects of interest, with Bayesian evidence showing substantial evidence for the null for a three-way interaction between age group, reference and baseline discounting (age group × other's preference × self baseline $km$ interaction: $b = 0.04$, 95% CI $= [-0.18, 0.26]$, $Z = 0.34$, $P = 0.73$, $BF_{01} = 3.73$; age group × self baseline $km$ interaction: $b = -0.06$, 95% CI $= [-0.20, 0.08]$, $Z = -0.88$, $P = 0.38$, $BF_{01} = 2.44$; other's preference × self baseline $km$ interaction: $b = 0.07$, 95% CI $= [-0.09$ $0.23]$, $Z = 0.84$, $P = 0.40$, $BF_{01} = 1.10$; main effect of self baseline $km$: $b = -0.02$, 95% CI $= [-0.13, 0.09]$, $Z = -0.32$, $P = 0.75$, $BF_{01} = 4.63$). Additionally, we re-ran these analyses to confirm that results remained the same accounting for the order of others' preferences (see Supplementary Note 1) and possible outliers (Figure S5, Tables S6, S7).

Finally, we examined whether people showed susceptibility to social influence in general, regardless of the type of preference they learnt about. We found people were generally influenced by other people, regardless of the type of influence: one-sample nonparametric $t$ tests showed that the signed $D_{KL}$ values were significantly different from zero for both impulsive (grand median across two age groups $= 0.12$, $W = 6832$, $Z = -3.57$, $r_{(152)} = 0.30$ [0.15 0.45], $P < 0.001$) and patient others (grand median across two age groups $= 0.37$, $W = 8624$, $Z = -7.67$, $r_{(152)} = 0.65$ [0.53 0.75], $P < 0.001$). We also observed that, on average, participants were more influenced by patient compared to impulsive others ($V = 2634$, $Z = -3.55$, $r_{(126)} = 0.31$ [0.15 0.47], $P < 0.001$). This finding aligns with the observation that participants reported feeling more similar to patient others compared to impulsive ones ($b = 1.20$, 95% CI $= [0.53$ $1.78]$, $Z = 3.62$, $P < 0.001$).

### Socio-affective traits explain variability in susceptibility to impulsive social influence amongst older adults

Finally, we examined how individual variations in socio-affective traits modulated people's susceptibility to social influence. For this purpose, we performed an exploratory factor analysis on the self-report questionnaires completed by participants (see Methods). This enabled us to identify latent patterns of behaviour measured across the questionnaires (e.g., affective empathy and emotional sensitivity scores were highly correlated, $r_{(150)}$ [95% CI] $= -0.65$ $[-0.73, -0.55]$, $P < 0.001$), facilitating both conceptual interpretation and statistical inference by reducing the number of comparisons. The factor analysis uncovered three distinguishable dimensions across the subscales of the questionnaires included (Fig. S6A). Factor 1 (Autistic & alexithymic traits) involved high loadings ( $> 0.40$) from measures related to autism, alexithymia, cognitive empathy and social apathy, Factor 2 (Psychopathic traits) encompassed high loadings ( $> 0.40$) from measures of psychopathic traits, and Factor 3 (Affective empathy & emotional motivation) included high loadings ( $> 0.40$) from measures of affective empathy and emotional motivation. Comparing the two age groups on these factors showed that no overall age-related difference was observed in the factor 'Autistic & alexithymic traits', which was only supported by anecdotal Bayesian evidence of no difference (young mean [SE] $= -0.12$ [0.11], older mean [SE] $= 0.11$ [0.10]; $W = 2501$, $Z = -1.42$, $r_{(150)} = 0.12$ [0.01, 0.26], $P = 0.155$, $BF_{01} = 1.96$). However, older people scored significantly lower in both the factor 'Psychopathic traits' (young mean [SE] $= 0.25$ [0.11], older mean [SE] $= -0.24$ [0.10]; $W = 3944$, $Z = -3.89$, $r_{(150)} = 0.32$ [0.17 0.46], $P < 0.001$) and the factor 'Affective empathy & emotional motivation' (young mean [SE] $= 0.25$ [0.11], older mean [SE] $= -0.24$ [0.09]; $W = 3833$, $Z = -3.48$, $r_{(150)} = 0.28$ [0.12, 0.44], $P < 0.001$).

We correlated the scores from each factor with people's tendency to socially conform to others (Fig. S6B). We found a significant positive correlation between impulsive $D_{KL}$ and the factor 'Affective empathy & emotional motivation' scores amongst older participants (Spearman: $r_{s(71)} = 0.29$ [0.06, 0.48], $P = 0.014$; FDR-corrected for three factor comparisons $P = 0.043$; Fig. 2B, Table S8), but not amongst young people ($r_{s(66)} = -0.13$ $[-0.36, 0.11]$, $P = 0.30$, $BF_{01} = 5.33$). Moreover, the association between the factor 'Affective empathy & emotional motivation' scores and impulsive social influence was significantly stronger in older adults than in young adults (independent $Z$-test after Fisher's $Z$-transformation; differences in correlation coefficients [95% CI] $= -0.41$ $[-0.72, -0.08]$, $Z = 2.45$, $P = 0.014$;

Table S9). There was no statistically significant correlation found between patient $D_{KL}$ and the factor 'Affective empathy & emotional motivation' scores in either older ($r_{s(66)} = -0.11$ $[-0.34, 0.13]$, $P = 0.36$, $BF_{01} = 3.25$) or young ($r_{s(69)} = -0.06$ $[-0.29$ $0.18]$, $P = 0.63$, $BF_{01} = 6.48$) participants. The findings collectively suggest a specific association between socio-affective traits and susceptibility to impulsive social influence among older adults. Older adults who were more susceptible to impulsive social influence also reported being more affectively empathetic and emotionally motivated.

Additionally, the factor 'Autistic & alexithymic traits' showed a significant positive correlation with patient $D_{KL}$ amongst older adults ($r_{s(66)} = 0.34$ [0.11, 0.54], $P = 0.004$; FDR-corrected for three factor comparisons $P = 0.013$), but not amongst young adults ($r_{s(69)} = -0.04$ $[-0.27, 0.20]$, $P = 0.76$, $BF_{01} = 7.15$). In addition, the correlation between scores on the factor 'Autistic & alexithymic traits' and susceptibility to patient social influence was significantly stronger in older adults compared to young adults (independent $Z$-test after Fisher's $Z$-transformation; differences in correlation coefficients [95% CI] $= -0.38$ $[-0.68, -0.05]$, $Z = -2.27$, $P = 0.023$; Table S9). No significant association was observed between impulsive $D_{KL}$ and the scores from the 'Autistic & alexithymic traits' in either older ($r_{s(71)} = -0.07$ $[-0.30, 0.16]$, $P = 0.54$, $BF_{01} = 6.59$) or young adults ($r_{s(66)} = 0.11$ $[-0.13, 0.34]$, $P = 0.35$, $BF_{01} = 4.90$). This suggests that older people with higher levels of autistic and alexithymic traits are more likely to be affected by patient social influence. Notably, Bayesian analysis also showed that psychopathic traits did not account for individual variations in susceptibility to both impulsive and patient social influence, in both young and older adults (all $BF_{01}s > 5.20$; Table S8). Moreover, all these results remained the same after removing outliers (Table S10).

### Discussion

People tend to alter their behaviours to imitate others once they become cognisant of their preferences. Using a delegated inter-temporal choice task and Bayesian computational models, we tested how young (aged 18–36) and older (aged 60–80) adults were susceptible to impulsive and patient social influence. We found that older adults were more affected by impulsive others compared to young adults. Furthermore, amongst the older adults, those more influenced by impulsive social influence reported higher levels of affective empathy and emotional motivation. This heightened susceptibility to social influence occurred despite both age groups being able to learn others' preferences, and despite no evidence of difference in their baseline temporal impulsivity.

Compared to young adults, we showed that older adults demonstrated a relatively greater susceptibility to social influence, particularly of impulsive others. Previous studies have suggested that older adults might be more sensitive to misinformation[20] and therefore preferences and information shared by other people. However, we show that this effect is specific to preferences considered impulsive, as older adults were relatively more swayed by impulsive others compared to young adults. Inconsistent findings have emerged from studies examining the influence of ageing on social conformity. Early studies using visual perceptual judgement tasks showed older adults demonstrated either increased[72] or decreased[73] susceptibility to social influence relative to young adults. However, another study using a collaborative delay discounting task observed no discernible difference in the susceptibility between the two age groups[74]. Notably, in this latter study, participants' choices were not incentivized and were unmatched, as young adults received course credit and older adults received $30 regardless of their decisions. Consequently, their choices may not have reflected true preferences[75]. We were able to fit detailed computational models of incentivized choices, and separately measure susceptibility to patient and impulsive influence. Another recent study using experience sampling showed that compared to young adults (aged 18–30 years), middle-aged (aged 31–59 years) and older adults (aged 60–80 years) were more likely to practise self-control when others were present enacting the desire, suggesting older people were less susceptible to social influence than young adults[76]. The distinct patterns between this study and ours may also be attributed to several factors: the differing composition of young and older

adult participants, different aspects of social conformity investigated, and different experimental approaches. Together these studies highlight the importance of considering the multidimensional aspect of social influence and utilising difference techniques to understand how susceptibility to social influence evolves over the course of life. It will be important for future work to examine dynamic fluctuations in susceptibility to social influence across the lifespan. Here and in other studies researchers have often focussed on cross-sectional samples for feasibility and increased power, yet longitudinal studies and those that include mid-life samples are crucial for enhancing our understanding of the process of social influence.

It is somewhat surprising that people shift their preferences to align with someone else when the shift could negatively impact their bonus payment. Social influence has been previously shown to operate in several different domains, including risk, and across different model-based and model-free analytical techniques[13–16,77–79]. An important next question is not how but why people shift; various explanations remain plausible. Participants might seek to learn about social norms[12] or gather information to reduce uncertainty[14]. These processes could be underpinned by more fundamental neural mechanisms, such as plasticity in the medial prefrontal cortex[13].

Another key finding was that younger adults did not show susceptibility to impulsive social influence, it neither made them more impulsive or more patient, instead they, on average, did not shift their preference. There are several explanations for this finding which could be probed in future research. For example, some studies have found that compared to adolescents, young adults are less susceptible to social influence across several domains (reviewed in ref. 80). Reduced social influence in young adults has been attributed to their reduced normative pressure to conform to others or their greater certainty about their own preferences, making them less likely to be influenced than adolescents[16]. It is interesting here that both young and older adults were susceptible to patient social influence, and that older adults who anecdotally may be expected to hold stronger belief certainty, were influenced in both domains. Again, together these studies highlight the importance of future work considering the whole lifespan.

Both theoretical accounts and empirical studies have shown that both adolescents and older adults display increased sensitivity to social rewards, such as rewards that help another person, compared to young adults[21,81–83]. Such a developmental trajectory might provide an explanation for why only older adults demonstrated increased susceptibility to social influence. The asymmetric social influence of impulsive others on young and older adults may reflect the observation that older people tend to have more polarised political views[84] and less flexible impressions of dissimilar others[85]. Importantly, we also discovered that the extent of such susceptibility was linked to their self-reported levels of emotional motivation, and this correlation was only found for older adults. Future studies could attempt to uncover the pharmacological basis of these effects. One study showed that the secretion of oxytocin following a social prime increased with advancing age[86] and oxytocin has been shown to foster social conformity[87–90] and enhance emotional sensitivity[91], suggesting a putative neuropeptide pathway.

Ageing is often associated with a decline in cognitive abilities, which can lead to poorer learning performance[23,92]. Contrary to expectations, our study showed the performances of learning about the others' preferences were similar between the two age groups. This intriguing finding dovetails with recent research indicating similar results in various facets of social learning. For example, in a study using a probabilistic reinforcement learning task, it was discovered that both young and older adults exhibited equivalent proficiency in learning what actions would benefit the anonymous other person. This finding suggests that the prosocial learning of older adults remains intact[24]. These findings also support the idea that social motivations progressively exert more influence on learning and decision-making as individuals age[93,94].

Although older adults showed no significant difference in learning accuracy, they did report lower confidence in their learning abilities, which can be seen as a judgement of metacognition. Studies of metacognition in other domains such as memory have reported that older adults may display

over-confidence[95]. However, in other domain such as visual perception, they may display under-confidence[96], suggesting that ageing may not be associated with global shifts in confidence. Notably, in our study, a confidence judgement was only provided at the end of the task rather than after each trial. Future studies could probe further whether older adults have insight into their greater influence by impulsive others for understanding whether and how such effects can be modified.

Future studies could also examine how much insight older adults have into their greater influence by impulsive others to understand whether and how such effects can be modified. There is also increasing empirical interest in the relationship between metacognition and mentalising, with possible computational and neural overlap between them[97–99]. In our task, another possible interpretation is that being influenced by others' beliefs may be related to one's theory of mind ability, given that the choices of others were inferred rather than directly observed. However, we found people displayed different susceptibility to impulsive and patient social influence. Even if theory of mind is indeed involved, it should theoretically be engaged in both scenarios. In addition, studies of older adults often suggest reduced or preserved theory of mind[100], whereas we found relatively enhanced susceptibility to social influence in older adults. Future studies could dissect and dissociate how and why metacognition, mentalising, and social influence drive any differences between older and young adults. Situating our experimental paradigm within the framework of metacognition or theory of mind could provide invaluable insights into how and why people's preferences are swayed by observing or inferring others' behaviours.

We also found that there was no significant difference in baseline temporal impulsivity between young and older adults. Studies of intertemporal preferences across the adult lifespan have shown mixed results[81]. Some have reported that older adults were more willing to wait for delayed offers[26,28,29], while others revealed an increased temporal impulsivity with age[27] or no difference in discounting preferences between young and older adults[30–32]. According to recent meta-analyses on this topic[33,34], there was no noticeable difference in intertemporal preferences between young (approximately 30 years old) and early older adults (around 70 years old), which is consistent with our findings here. No significant difference in baseline temporal impulsivity between the two age groups provides a solid foundation for comparing their susceptibility to social influence. However, in follow-up analyses, we also showed that controlling for baseline impulsivity did not alter our findings.

## Limitations

In addition to these novel findings, there are also limitations. Firstly, we focussed on a specific type of social influence related to economic preferences. Future research could examine a broader range of social influences that may differ between young and older adults. Secondly, our study was cross-sectional by design. Longitudinal studies are needed incorporating mid-life samples to understand how susceptibility to social influence evolves throughout adulthood and account for possible non-linear changes. Third, we used validated computational models of social influence. However, these models do not incorporate social-cognitive aspects such as theory of mind that may account for why people shift their preferences and behaviours and individual differences between people. Finally, the abstract nature of the experimental design may have missed some of the complexities inherent in real-world social influence scenarios[76]. Future research could consider examining social influence in everyday experiences.

## Conclusion

Our findings provide evidence that older adults, in contrast to young adults, were more susceptible to the influence of impulsive others, and the degree of this susceptibility was associated with their self-reported levels of emotional motivation. This observation holds true even though older adults demonstrated no statistical difference in ability to learn others' preferences, and there were no significant differences in their baseline impulsivity. We also found that age group differences in susceptibility were not explained by variations in general IQ or executive function. Together, these findings may

**Article**

have significant implications for understanding susceptibility to social influence, how age differences may affect susceptibility to misinformation, and the challenges and opportunities of an ageing population.

## Data availability
Data are available at https://osf.io/zgb5v/.

## Code availability
Code for modelling and analysis is available at https://osf.io/zgb5v/.

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

## Acknowledgements

P.L.L. was supported by a Medical Research Council Fellowship (MR/P014097/1 and MR/P014097/2), a Jacobs Foundation Research Fellowship, a Sir Henry Dale Fellowship funded by the Wellcome Trust and the Royal Society (223264/Z/21/Z), a UKRI/EPSRC Frontier Science Guarantee (ERC Starting Grant Replacement Funding, EP/X020215/1) and a Leverhulme Prize from the Leverhulme Trust (PLP-2021-196). S.G.M. was supported by the National Institute for Healthcare Research (NIHR) Oxford Biomedical Research Centre (BRC). Z.S. was supported by the Government Scholarship of Overseas Study funded by the Ministry of Education in Taiwan. The funders had no role in study design, data collection and analysis, decision to publish or preparation of the manuscript. We would also like to thank Ayat Abdurahman, Daniel Drew and Luca Hargitai for assistance with data collection. We would also like to thank Andrea Reiter and Michael Moutoussis for their useful advice regarding the Bayesian computational modelling.

## Author contributions

Conceptualization: P.L.L., M.M.G., M.A.J.A., S.G.M., L.T., J.H.B. Methodology: P.L.L., M.M.G., S.G.M., L.T., J.H.B. Investigation: P.L.L. Formal analysis: Z.S., M.M.G., L.Z., T.A.V., L.T. Writing – Original Draft: Z.S., P.L.L. Writing – Review & Editing: P.L.L., M.M.G., M.A.J.A., S.G.M., L.T., J.H.B., L.Z., T.A.V., M.H., Z.S. Funding Acquisition: M.H., P.L.L. Supervision: T.A.V., P.L.L.

## Competing interest

The authors declare the following competing interests: P.L.L. is an Editorial Board Member for *Communications Psychology*, but was not involved in the editorial review of, nor the decision to publish this article.
