## [transparent peer review · Communications Psychology]

Decision letter and referee reports: first round

Dear Mr Su,

Your manuscript titled "Older adults are relatively more susceptible to impulsive social influence than young adults" has now been seen by our reviewers, whose comments appear below. In light of their advice I am delighted to say that we are happy, in principle, to publish a suitably revised version in *Communications Psychology*.

We therefore invite you to revise your paper one last time to address the remaining concerns of our reviewers and a list of editorial requests. The limitations section can serve to mention remaining criticisms from the reviewers. At the same time we ask that you edit your manuscript to comply with our format requirements and to maximise the accessibility and therefore the impact of your work.

EDITORIAL REQUESTS:

SUBMISSION INFORMATION:

OPEN ACCESS:

* **DATA AVAILABILITY:**

Please use the following link to submit the above items:
(link Redacted)

** This url links to your confidential home page and associated information about manuscripts you may have submitted or be reviewing for us. If you wish to forward this

email to co-authors, please delete the link to your homepage first **

Best regards,

Jennifer Bellingtier

Jennifer Bellingtier, PhD
Senior Editor
Communications Psychology

REVIEWERS' EXPERTISE:

Reviewer #1 development/decision-making

Reviewer #2 social decision making/computational models

REVIEWERS' COMMENTS:

Reviewer #1 (Remarks to the Author):

Thank you for the opportunity to re-review this manuscript. The authors have been very responsive to my (and other reviewers') comments, and I only have one remaining concern. The current title, "Older adults are relatively more susceptible to impulsive social influence than young adults," still implies that older adults are the age group deviating from the norm when, in fact, it is the younger adults who are doing so. A simple flip, "Younger adults are relatively less susceptible to impulsive social influence than older adults," emphasizes that younger adults are the group who are deviating.

Reviewer #2 (Remarks to the Author):

I appreciate the time and effort that the authors have dedicated to this comprehensive revision. The manuscript is much more comprehensible now and the additional analysis strengthen the results. The PPC that I recommended was conducted very carefully and the

findings support the modeling results.

What I did not find satisfactorily addressed the conceptual issue of involvement of Theory of Mind. The authors argue that they did not manipulate ToM levels and that ToM does not explain the differences effects of learning about other's preferences observed in younger and older adults. But that was not my argument in the first place. The task itself does not feature direct social influence (despite the claim in previous publications): at no point during the task, the participant is shown the actual choices of another person (which happens to be not real after all). Rather, the participant makes choices *for* another agent and learns about the agents delay preferences only through feedback. It really doesn't get more indirect than this! Rather it seems pretty obvious to me that in this task the participant learns to construct a mental model of the other agent's delay preference and this self-evident fact puts this task in the ToM domain (which btw do not always have to feature different levels). Of course, the finding that self choices are modulated by the mental model learned for the other agent can be labeled as "social influence", but this is really different from what the field normally understands as social influence, which features observing the behavior of others and subsequently adapting one's own behavior accordingly. Here, it's all the "head" of the participant, no social behavior (i.e. choices of others) are ever observed.

I am not saying that the findings of this paper are uninteresting and not worth publishing. But I do think that re-framing the task in terms of ToM would open up a new set of fascinating questions of how we interact with our mental models, and these questions remain unexpressed and unanswered here.

Another important aspect that was not properly addressed in the revision was the question about the cognitive mechanisms that are involved in modulating self choices after learning about others, and this shortcoming persists regardless of the whether the task is framed as "social influence" or "mentalizing": through model-free and model-informed analyses the authors demonstrate the shift in own delay preference, but they do not provide any explanation of how, let alone why, the modulation is realized psychologically and computationally.

Author Responses: first round.

Older adults are relatively more susceptible to impulsive social influence than young adults

Reviewer #1 (Remarks to the Author):

Thank you for the opportunity to review this paper. The authors use a social conformity task in younger and older adults and find that other than older adults, younger adults copy patient behaviour more than impulsive behaviour. The study is well-done, uses sophisticated modeling, and the results are fairly interesting, (although, I speculate, perhaps particularly so for a more specialized readership than this journal attracts?)

Response: We really appreciate your positive feedback on our manuscript. Thank you for taking the time to review our work and for your insightful comments that have contributed to improving our manuscript.

R1.1: My personal conviction in peer review is that an evaluation of novelty and impact should be left to the editor alone, but I want to point out that this is a cross-sectional study on a topic that has been researched before using cross-sectional approaches, including recent ones with high external validity:

Castrellon, J. J., Zald, D. H., Samanez-Larkin, G. R., & Seaman, K. L. (2024). Adult age-related differences in susceptibility to social conformity pressures in self-control over daily desires. *Psychology and Aging*, 39(1), 102–112

Some of them are discussed in the discussion section, however, one very related one (which might raise concerns about novelty of article at hand) is not referenced to:

Bixter, M. T., & Rogers, W. A. (2019). Age-related differences in delay discounting: Immediate reward, reward magnitude, and social influence. *Journal of Behavioral Decision Making*, 32(4), 471–484. <https://doi.org/10.1002/bdm.2124>

There are also plenty of studies looking at delay discounting in older vs. younger adults. I think longitudinal data, or including a middle-aged sample might be steps forward the authors have not taken here.

Response: Thank you for highlighting other important work related to this interesting research topic. Despite this related work, we believe our study still carries significant novelty in multiple domains.

Castrellon and colleagues used experience sampling surveys to examine the effect of the presence of others on self-control in three distinct age groups. They showed that compared to young adults (aged 18-30 years), middle-aged (aged 31-59 years) and older adults (aged 60-80 years) were more likely to practise self-control when there were others present enacting the desire. However, there are several critical differences between our study and theirs, supporting empirical novelty in our results. Firstly, the conceptualisation of socially conforming to others differs between the two studies. In the study by Castrellon et al. (2024), social conformity was defined as the inability to resist desire when there were others present enacting the same desire, emphasising external behaviours rather than underlying preferences. On the other hand, our study tests how people's underlying preferences are influenced by observing others' economic preferences, and social conformity was defined as participants' preferences shifted towards those of others. Given the different pattern of results in the two studies, our findings highlight the importance of different dimensions of social

conformity and perhaps that people might change control over their actions without changing their preferences.

Secondly, the study design is distinctive between the two manuscripts. Castellon et al. (2024) took advantage of the experience-sampling technique to achieve external validity. However, this approach involved a trade-off as the heterogeneity of scenarios—such as varying types of desires, numbers of others present, and types of social interactions—was not controlled. We adopted a laboratory design that, while potentially limiting external validity, enabled us to apply precise and advanced computational modelling. This approach helped us reveal the latent factors driving susceptibility to patient and impulsive influence.

Finally, the composition of participants differs. In Castellon et al.'s study (2024), across the three samples included in the analysis, there were 13 older adults (aged 60-80 years) out of total 157 participants (8.3%), whereas we had 78 older adults (aged 60-80 years) out of total 154 participants (50.6%). The larger sample of older adults aids us in drawing stronger conclusions about young and older adults as distinct groups, although we agree future work would benefit from also incorporating a longitudinal or large lifespan approach. We have now added this point as a future direction and limitation in our discussion.

Regarding Bixter and Rogers (2019) we already briefly discussed their paper in our original manuscript, but we inadvertently omitted the reference during editing. We have now added the reference and again highlighted why our study provides novel contributions. In particular, in Bixter and Rogers participants choices were not incentivized directly and differed between young and older adults (young adults received course credit and older adults received a fixed \$30 regardless of their choices). Consequently, their choices may not reflect their true preferences⁴⁹. Secondly, because we independently measured patient and impulsive social influence through multiple trials we were able to fit detailed computational models that can be generalised to different contexts and populations and can be more sensitive when uncovering latent preferences. Finally, Bixter and colleagues asked participants to discuss their preferences in matched dyads that were video recorded. We show that simply learning about another's preference differently impacts young and older adults' susceptibility to social influence. We again believe these important differences highlight strengths and weakness of both approaches in this emerging research field.

Discussion:

However, another study using a collaborative delay discounting task observed no discernible difference in susceptibility between the two age groups⁴⁴. Notably, in this latter study, participants' choices were not incentivized and were unmatched, as young adults received course credit and older adults received \$30 regardless of their decisions. Consequently, their choices may not have reflected true preferences⁴⁵. We were able to fit detailed computational models of incentivized choices, and separately measure susceptibility to patient and impulsive influence. Another recent study using experience sampling showed that compared to young adults (aged 18-30 years), middle-aged (aged 31-59 years) and older adults (aged 60-80 years) were more likely to practise self-control when others were present enacting the desire, suggesting older people were less susceptible to social influence than young adults. The distinct patterns between this study and ours may

also be attributed to several factors: the differing composition of young and older adult participants, different aspects of social conformity investigated, and different experimental approaches. Together these studies highlight the importance of considering the multidimensional aspect of social influence and utilising difference techniques to understand how susceptibility to social influence evolves over the course of life. It will be important for future work to examine dynamic fluctuations in susceptibility to social influence across the lifespan. Here and in other studies researchers have often focussed on cross-sectional samples for feasibility and increased power, yet longitudinal studies and those that include mid-life samples are crucial for enhancing our understanding of the process of social influence.

R1.2: I have another major issue: I think the results are presented and discussed in a perhaps slightly misleading way – what seems to really happen here is that younger adults show specificity in their conformity, namely that they are much less likely to adopt behavior of an impulsive other than of a patient other, whereas this difference is not significant in older adults. That is, the interaction effect is driven by an effect in YA that is not significant in OA. Put differently, the really interesting bar is the YA one on the left hand side – YA virtually show no contagion effect towards impulsive others. Why is that? I think this is the main finding and should be discussed accordingly, reflected in the title etc. I name some examples in the following, but it affects the whole intro and discussion section, I think: interpretations like

“Here we show in an incentivized and controlled task accounting for baseline discount preferences that older adults are relatively more influenced by the preferences of impulsive others.” seem misleading as they suggest that there is something special about OA, where as the interesting differentiation happens in the YA.

“Such a developmental trajectory might provide an explanation for why only older adults demonstrated increased susceptibility to social influence.” This suggests a main effect of age on susceptibility which is not reported (and if it existed, again driven by the reduced susceptibility of YA towards impulsive others.)

In this vein, I think a title like “ Younger, but not older adults are less susceptible to impulsive social influence than to patient social influence” would do the findings more justice (I agree it is perhaps less shiny). Don’t get me wrong, I think it would be really interesting to understand/ discuss the reduced effect of impulsive others on behavior in YA, but this is not how the paper is framed now. What do the authors think where this comes from?

Response: Thank you for this feedback on our presentation of the results. We agree that our manuscript could be edited to describe our results without any ambiguity. We believe it is important to highlight the relative difference between young and older adults in impulsive social influence as a central result for several reasons. First, in our design susceptibility to social influence is continuous so the difference between young and older adults, who we carefully matched, is relative. Second, we used two distinct groups to maximise power. Comparing these two groups shows a relative difference in impulsive social influence. Third, susceptibility to impulsive and patient social influence may operate through separate mechanisms and therefore the comparison on impulsive influence between young and older adults is valid. In a new analysis (also in response to **R1.6**) we found that individual differences in susceptibility to impulsive social influence were correlated with traits related to affective empathy and

emotional motivation. Conversely, individual variations in susceptibility to patient social influence correlated with autistic and alexithymic traits.

We believe that emphasising the relative difference between young and older adults could be acknowledged more clearly in the key conclusions of our manuscript. We have therefore now revised our title to '**Older adults are relatively more susceptible to impulsive social influence than young adults**'. In addition, the relevant parts in the abstract and throughout the manuscript have also been updated.

Abstract:

We used the signed Kullback-Leibler divergence to quantify the magnitude and direction of social influence. We found that, compared to young adults, older adults were relatively more susceptible to impulsive social influence.

Discussion:

Another key finding was that younger adults did not show susceptibility to impulsive social influence, it neither made them more impulsive or more patient, instead they, on average, did not shift their preference. There are several explanations for this finding which could be probed in future research. For example, some studies have found that compared to adolescents, young adults are less susceptible to social influence across several domains (reviewed in⁵⁰). Reduced social influence in young adults has been attributed to their reduced normative pressure to conform to others or their greater certainty about their own preferences, making them less likely to be influenced than adolescents¹⁶. It is interesting here that both young and older adults were susceptible to patient social influence, and that older adults who anecdotally may be expected to hold stronger belief certainty, were influenced in both domains. Again, together these studies highlight the importance of future work considering the whole lifespan.

R1.3: If I understood correctly, there are more free parameters in the winning model. Were there age differences in KU as well? This is plausible as older adults have been demonstrated to show more variability in a series of studies before. If there were, how did they relate to the age differences reported?

Response: We apologise for any ambiguity in our modelling results. In the winning model (i.e., the KU model without noise parameter), there are only two free parameters, the mean (k_m) and standard deviation (k_u) of the discounting distribution per participant. No differences were observed in the mean (young group mean [SE] = -4.79 [0.22], older group mean [SE] = -5.16 [0.25]; independent Wilcoxon signed-rank test: $W = 3243$, $Z = -1.01$, $r(152) = 0.08$ [0.005 0.24], $p = 0.314$, $BF_{01} = 3.47$) or standard deviation (young group mean [SE] = 1.37 [0.06], older group mean [SE] = 1.47 [0.06]; $W = 2481$, $Z = -1.74$, $r(152) = 0.14$ [0.009 0.31], $p = 0.081$, $BF_{01} = 2.31$) of the discounting distribution at baseline across the two age groups. We are pleased to include the figures below in our supplement to better illustrate the results (**Figure S3**).

Figure S3. Baseline temporal impulsivity and preference uncertainty do not significantly differ with age. No statistically significant differences were noted in either the mean (independent Wilcoxon signed-rank test: $W = 3243$, $Z = -1.01$, $r(152) = 0.08$ [0.005 0.24], $p = 0.314$, $BF_{01} = 3.47$) or the standard deviation ($W = 2481$, $Z = -1.74$, $r(152) = 0.14$ [0.009 0.31], $p = 0.081$, $BF_{01} = 2.31$) of the discounting distribution at the baseline between two age groups.

R1.4: Thank you for reporting the recovery analysis. Can you show the same group differences on simulated data as you do on the empirical data (posterior predictive checks)?

Response: Thank you for suggesting performing posterior predictive checks. We have now used posterior predictive checks to evaluate the extent to which the posterior estimation of our winning model accurately replicates the key features of participant behaviour (e.g., their performances of learning about others' preferences). For this purpose, we used a post-hoc absolute-fit approach (Zhang et al., 2020), which factored in participants' actual decisions and option pairs, to generate predictions using the entire set of posterior Markov chain Monte Carlo (MCMC) samples of the winning model. To be specific, we let the winning model generate synthetic decisions repeatedly, matching the number of MCMC samples (i.e., 8000 times) for each trial and each participant, with the individual-level posterior parameters obtained through model estimation. Then, we analysed the synthetic data using the same methods as those applied to the actual data, using a linear mixed-effects model (LMM). This LMM incorporated fixed effects of age group (older vs young), other's preference (patient vs impulsive), and their interaction, along with a random subject-level intercept.

We found that the posterior prediction accurately replicated the key patterns in our behavioural data (**Figure S1**): learning the preferences of patient others was more accurate compared to impulsive ones (significant main effect of other's preference: $b = 0.03$, 95% CI = [0.02 0.05], $Z = 4.35$, $P < 0.001$), whereas there was no significant difference in learning performance between the two age groups (main effect of age group: $b = -0.01$, 95% CI = [-0.03 0.01], $Z = -1.04$, $P = 0.30$, $BF_{01} = 2.47$; age group x other's preference interaction: $b = -0.002$, 95% CI = [-0.02 0.02], $Z = -0.21$, $P = 0.83$, $BF_{01} = 6.21$). This indicates that our winning model could indeed capture our empirical findings accurately, verifying its validity.

Figure S1. Posterior predictive checks of the winning KU model. Comparison of simulated learning accuracy shows no statistically significant difference in learning performance of the others' preferences between the two age groups (no main effect of age group: $b = -0.01$, 95% CI = [-0.03 0.01], $Z = -1.04$, $P = 0.30$, $BF_{01} = 2.47$). In addition, both simulated young and older participants demonstrated better learning of the preferences of patient others compared to impulsive ones (significant main effect of other's preference: $b = 0.03$, 95% CI = [0.02 0.05], $Z = 4.35$, $P < 0.001$). Large circles with border lines indicate the mean, error bars represent the standard error of the mean, dots show raw data, and asterisks denote the significant main effect of other's preference based on the linear mixed-effects model. Note that the vertical axis begins at 50%, representing the chance level. *** $P < 0.001$; ns: not significant. Red dots are the means of actual data.

R1.5: Did the order of presentation influence the results of interest? In my experience with social tasks [end of comment missing]

Response: Thank you for the suggestion to examine the effect of the order of others' preferences on our results or interest, the signed KL divergence. We examined the following two linear mixed-effects models (LMMs) that included an additional fixed effect of the order of

other's preferences (patient first vs impulsive first) without interacting with other terms, as a control variable:

LMM1: Signed KL divergence - Group * Preference + **Order** + (1|ID)

LMM2: Signed KL divergence - Group * Preference * Self baseline impulsivity + **Order** + (1|ID)

Controlling for the order of other's preferences **did not** change any of our main results, without (LMM1: main effect of **order of others**: $b = 0.07$, 95% CI = [-0.11 0.25], $Z = 0.75$, $P = 0.46$, $BF_{01} = 5.75$; significant main effect of age group: $b = 0.47$, 95% CI = [0.22 0.73], $Z = 3.6$, $P < 0.001$; significant main effect of other's preference: $b = 0.57$, 95% CI = [0.31 0.83], $Z = 4.33$, $P < 0.001$) or with (LMM2: main effect of **order of others**: $b = 0.07$, 95% CI = [-0.11 0.26], $Z = 0.79$, $P = 0.43$, $BF_{01} = 6.07$; significant main effect of age group: $b = 0.45$, 95% CI = [0.19 0.71], $Z = 3.38$, $P < 0.001$; significant main effect of other's preference: $b = 0.54$, 95% CI = [0.28 0.81], $Z = 3.99$, $P < 0.001$; significant interaction between age group and other's preference: $b = -0.53$, 95% CI = [-0.91 -0.16], $Z = -2.78$, $P = 0.005$) self baseline impulsivity as covariates.

We have added these results to our results section, thank you for the suggestion.

R1.6: I get a little bit suspicious if correlations with only one subscale are reported in non-preregistered studies. What are the other subscales and why is there more theoretical justification for testing this specific one? Were there other questionnaires assessed in this study and might there be multiple testing issues? Thank you for transparent reporting!

Response: Thank you for raising the issue of correlations of multiple self-report questionnaires. In the study, we chose to focus on emotional motivation as theoretically, we hypothesised that this subscale may most accurately explain the variability in being influenced by others, as we highlight in the manuscript. We were also mindful to limit the number of multiple comparisons to balance between type 1 and type 2 errors. In focussing on this subscale, we certainly did not want to be untransparent, as in most large studies in the lab we indeed collect additional measures for completeness, but had no strong theoretical reason to focus on those instead.

To ensure we could include all measures and further highlight the robustness of the emotion motivation association, we have now taken a data-driven rather than theory-driven approach and conducted an exploratory factor analysis including all collected measures. This analysis revealed a three-factor latent structure, where the third factor ('Affective empathy & emotional motivation') included the emotional sensitivity subscale from the AMI and the affective empathy subscale from the QCAE. This result aligns with our original hypothesis suggesting a robust connection between emotional motivation and affective empathy. Using scores from the factor 'Affective empathy & emotional motivation' also replicated our previous findings: the significant association between traits related to emotional motivation and social influence was only observed amongst older people when the influence was impulsive, and older people who were more emotionally motivated (and affectively empathetic) showed greater susceptibility to impulsive social influence (see changes to the text in the manuscript below).

In addition, this analysis also revealed a new result: a significantly positive correlation between the factor 'Autistic & alexithymic traits' scores and people's susceptibility to patient social

influence amongst older participants, which suggests older people with higher levels of autistic and alexithymic traits are more influenced by patient social influence. This correlation was not found amongst young people or for impulsive social influence. We have added these new analyses to our results section:

Results:

Finally, we examined how individual variations in socio-affective traits modulated people's susceptibility to social influence. For this purpose, we performed an exploratory factor analysis on the self-report questionnaires completed by participants (see Methods). This enabled us to identify latent patterns of behaviour measured across the questionnaires (e.g., affective empathy and emotional sensitivity scores were highly correlated, $r_{(150)} = -0.65$), facilitating both conceptual interpretation and statistical inference by reducing the number of comparisons. The factor analysis uncovered three distinguishable dimensions across the subscales of the questionnaires included (**Figure S6A**). Factor 1 (Autistic & alexithymic traits) involved high loadings (> 0.40) from measures related to autism, alexithymia, cognitive empathy and social apathy, Factor 2 (Psychopathic traits) encompassed high loadings (> 0.40) from measures of psychopathic traits, and Factor 3 (Affective empathy & emotional motivation) included high loadings (> 0.40) from measures of affective empathy and emotional motivation. Comparing the two age groups on these factors showed that no overall age-related difference was observed in the factor 'Autistic & alexithymic traits' (young mean [SE] = -0.12 [0.11], older mean [SE] = 0.11 [0.10]; $W = 2501$, $Z = -1.42$, $r_{(150)} = 0.12$ [0.01 0.26], $P = 0.155$, $BF_{01} = 1.96$). However, older people scored significantly lower in both the factor 'Psychopathic traits' (young mean [SE] = 0.25 [0.11], older mean [SE] = -0.24 [0.10]; $W = 3944$, $Z = -3.89$, $r_{(150)} = 0.32$ [0.17 0.46], $P < 0.001$) and the factor 'Affective empathy & emotional motivation' (young mean [SE] = 0.25 [0.11], older mean [SE] = -0.24 [0.09]; $W = 3833$, $Z = -3.48$, $r_{(150)} = 0.28$ [0.12 0.44], $P < 0.001$).

We correlated the scores from each factor with people's tendency to socially conform to others (**Figure S6B**). We found a significant positive correlation between impulsive $_{KL}$ and the factor 'Affective empathy & emotional motivation' scores amongst older participants (Spearman: $r_{s(71)} = 0.29$ [0.06 0.48], $P = 0.014$; FDR-corrected for three factor comparisons $P = 0.043$), but not amongst young people ($r_{s(66)} = -0.13$ [-0.36 0.11], $P = 0.30$, $BF_{01} = 5.33$). Moreover, the association between the factor 'Affective empathy & emotional motivation' scores and impulsive social influence was significantly stronger in older adults than in young adults ($Z = 2.45$, $P = 0.014$). There was no statistically significant correlation found between patient $_{KL}$ and the factor 'Affective empathy & emotional motivation' scores in either older ($r_{s(66)} = -0.11$ [-0.34 0.13], $P = 0.36$, $BF_{01} = 3.25$) or young ($r_{s(69)} = -0.06$ [-0.29 0.18], $P = 0.63$, $BF_{01} = 6.48$) participants. The findings collectively suggest a specific association between socio-affective traits and susceptibility to impulsive social influence among older adults. Older adults who are more susceptible to impulsive social influence also report being more affectively empathetic and emotionally motivated.

Additionally, the factor 'Autistic & alexithymic traits' showed a significant positive correlation with patient $_{KL}$ amongst older adults ($r_{s(66)} = 0.34$ [0.11 0.54], $P = 0.004$;

FDR-corrected for three factor comparisons $P = 0.013$), but not amongst young adults ($r_{S(69)} = -0.04 [-0.27 \ 0.20]$, $P = 0.76$, $BF_{01} = 7.15$). No significant association was observed between impulsive I" and the scores from the 'Autistic & alexithymic traits' in either older ($r_{S(71)} = -0.07 [-0.30 \ 0.16]$, $P = 0.54$, $BF_{01} = 6.59$) or young adults ($r_{S(66)} = 0.11 [-0.13 \ 0.34]$, $P = 0.35$, $BF_{01} = 4.90$). This suggests that older people with higher levels of autistic and alexithymic traits are more likely to be affected by patient social influence. Notably, Bayesian analysis also showed that psychopathic traits did not account for individual variations in susceptibility to both impulsive and patient social influence, in both young and older adults (all $BF_{01}s > 5.20$; see **Table S8**). Moreover, all these results remained the same after removing outliers (**Table S9**).

Figure S6. Individual variations in susceptibility to social influence amongst older adults are related to socio-affective traits. (A) Factor loadings for each subscale of the questionnaires completed. Exploratory factor analysis uncovered three distinct dimensions related to social affective cognition and behaviour: autistic & alexithymic

traits, psychopathic traits, and affective empathy & emotional motivation. Note: the loadings displayed here are absolute values. AQ: Autism Quotient; AMI: Apathy-Motivation Index; TAS: Toronto Alexithymia Scale; QCAE: Questionnaire of Cognitive and Affective Empathy; SRP: Self-Report Psychopathy scale. (B) The factors ‘Autistic & alexithymic traits’ and ‘Affective empathy & emotional motivation’ related to susceptibility to social influence in older adults, depending on the nature of social influence. Older people with higher levels of autistic and alexithymic traits are more susceptible to patient social influence, while those who are more affectively empathetic and emotionally motivated display greater susceptibility to impulsive social influence.

Table S8. Correlations between the factors and signed KL divergence (D_{KL})

			Autistic & alexithymic traits	Psychopathic traits	Affective empathy & emotional motivation
Young	Impulsive	$r_{S(66)}$	0.11 [-0.13 0.34]	0.01[-0.23 0.25]	-0.13 [-0.36 0.11]
	D_{KL}	BF_{01}	4.90	6.85	5.33
	Patient D_{KL}	$r_{S(69)}$	-0.04 [-0.27 0.20]	-0.07 [-0.30 0.17]	-0.06 [-0.29 0.18]
		BF_{01}	7.15	6.40	6.48
Older	Impulsive	$r_{S(71)}$	-0.07 [-0.30 0.16]	0.03[-0.20 0.26]	0.29 (0.06 0.48)*
	D_{KL}	BF_{01}	6.59	6.51	0.25
	Patient D_{KL}	$r_{S(66)}$	0.34 (0.11 0.54)**	0.11[-0.13 0.34]	-0.11 [0.34 0.13]
		BF_{01}	0.22	5.22	3.25

Note. $r_{S(df)}$: Spearman’s Rho correlation coefficients with degrees of freedom; 95% confidence intervals are indicated in square brackets. BF_{01} indicates the strength of evidence for the null hypothesis. * $P < 0.05$, ** $P < 0.01$.

Methods:

Exploratory factor analysis

We performed an exploratory factor analysis on the questionnaire subscales using the ‘fa’ function (from the {psych} v2.4.3 package) in R v4.2.1. We incorporated all the subscales from the Autism Quotient (AQ), Apathy-Motivation Index (AMI), Toronto Alexithymia Scale (TAS), Self-Report Psychopathy scale (SRP), and the Questionnaire of Cognitive and Affective Empathy (QCAE). To extract factor loadings, we used maximum likelihood estimation with an oblimin rotation. Regarding the determination of the number of factors (using the ‘fa.parallel’ and ‘vss’ functions from the {psych} v2.4.3 package), the Kaiser rule (eigenvalue > 1) pointed toward a 2-factor solution, the very simple structure (VSS) complexity 2 criterion implicated a 3-factor solution, examination of the scree plot indicated a 3-factor solution, and parallel analysis suggested a 4-factor solution. After weighing parsimony and interpretability of the latent structure, we settled on the 3-factor solution. This 3-factor latent structure explained 50.55% of the variance in the measures, with moderate correlations with each other (highest $r = 0.25$). Individual scores for each factor were calculated using Thurstone’s at the participant level. These scores were subsequently correlated with the signed KL divergence using Spearman’s Rho correlation coefficients.

R1.7: “No participant reported disbelief that these choices were from other people” What does this mean? This line of argumentation sounds a little bit like “absence of an effect is interpreted as no effect” The How was this assessed? Might age differences in the credibility of the coverstory explain age differences in credibility?

Response: Thank you for providing the opportunity for us to clarify our social manipulation. We excluded all participants with backgrounds in psychology or related disciplines, who might have guessed the actual purposes and objectives of the study. In addition, upon review of our experiment notes, it indicates that no participant raised questions regarding the authenticity of their interaction with others. As no one raised the issue, there was no age group difference in terms of their belief in our social manipulation. We have added further clarifications to our methods.

Methods:

No participant reported to the experimenter that they disbelieved the choices they observed were from other people. We further probed whether they had any disbelief in a post-study survey by asking if they had any questions or concerns about the task they completed. Both checks further demonstrated the validity of our task.

R1.8: Can you give more details about the stats model, e.g. how was the random effects structure in the mixed models set up?

Response: Thank you for your query. Linear mixed-effects models (LMMs) were used to predict participants’ learning accuracy, signed KL divergence, and task-specific questionnaires. These models included fixed effects of age group (older vs young), other’s preference (patient vs impulsive), and their interaction, along with a random subject-level intercept. An additional analysis of signed KL divergence also incorporated participants’ baseline temporal impulsivity (km; continuous covariates, centred around the grand mean) and its interaction with age groups and other’s preferences (including the three-way interaction) as fixed terms. In addition, control analyses of signed KL divergence included the standardised IQ scores from the WTAR as a fixed term, without interacting with the other terms (age group, other’s preference, or self baseline impulsivity). Please see the original manuscript page 30, lines 651-663. We have also now included the formula forms of statistical models in our supplement to improve readability:

Supplementary information:

The models were set up as follows:

*LMM1: Accuracy ~ Group * Preference + (1|ID)*

*LMM2: Questionnaires (confidence/similarity) ~ Group * Preference + (1|ID)*

*LMM3a: Signed KL divergence ~ Group * Preference + (1|ID)*

*LMM3b: Signed KL divergence ~ Group * Preference * Self baseline impulsivity + (1|ID)*

*LMM3c: Signed KL divergence ~ Group * Preference + IQ + (1|ID)*

*LMM3d: Signed KL divergence ~ Group * Preference * Self baseline impulsivity + IQ + (1|ID)*

Minor:

R1.9: I think the paragraph starting at line 232 should be the first paragraph of the results section – I find it more logical (and conventional) to start with the main effects and then move forward to the interactions that modulate the main effects (i.e. to your main finding of the age x preference interaction).

Response: Thank you for suggesting a different format to present the results. We chose to present our results in this manner to first showcase the key findings of interest and subsequently delve into the validation work.

R1.10: Does the main effect of age in line 143 (main effect $b = -0.01$, 95% CI = $[-0.04 \ 0.01]$... refer to accuracy on all trials or only on the patient trials? This was not clear from the text and Figure 2B suggests an age main effect (what do the error bars represent?!). I recommend reporting the age comparison along with the binomial test against chance.

Response: We apologise for any confusion in our reporting of the results. The non-significant main effect of age group on accuracy was computed across all trials, including both impulsive and patient trials.

In the original **Figure 1B** (which shows the results of learning accuracy, Figure 2B shows the modelling of KL divergence), the error bars represent the standard errors of the mean (please see the legend of Fig. 1B in the original manuscript page 7, lines 120-123). As shown in the figure, the error bars for young and older adults overlap on both impulsive and patient trials, suggesting that the main effect of age group was not statistically significantly different, consistent with the results from the LMM. However, we noted there was no strong Bayesian evidence for the null effect of this analysis ($BF_{01} = 1.56$).

Thank you for suggesting another way to present the results. We would like to start the results by emphasising that participants from both age groups could accurately learn others' preferences, and then test the age group differences in learning performances. We have updated the relevant paragraph to make this point more prominent:

Results:

Older and young adults can both learn others' preferences accurately

*To validate participants' ability to complete the task, we first examined whether they were able to learn the preferences of the other agents with different discounting preferences significantly above chance (50%) (see **Figure 1B**).*

R1.11: Line 620 "This method is more reliable than information criteria that are solely on point-estimates, such as the Akaike information criterion (AIC) and the Bayesian information criterion (BIC)." This needs a reference or any other form of evidence?

Response: Thank you for raising this point, and we have added the following references to the manuscript detailing the reliability of this method:

Methods:

Model comparison and parameter recovery

This method is more reliable than [...] Akaike information criterion (AIC) and the Bayesian information criterion (BIC)^{87, 89}.

Reviewer #2 (Remarks to the Author):

This paper administers a social delay discounting task first reported by Moutoussis et al. in PLoS CB to younger and older adults to determine through computational modeling whether older adults are more or less susceptible to social influence. Specifically, participants alternated between blocks, in which they had to complete 50 trials of delay discounting questions for their own preference, or they had to learn the preference of two other simulated agent (disguised as a previous participant) through feedback. These agent were simulated as more or less impulsive than the participant. In their primary finding the authors demonstrate that older adults were more susceptible to social influence than younger adults. The effect was amplified in older adults with high emotional motivation.

Several aspects of this study left a positive impression with me. The construction of the options in each trial of the self blocks combined a fixed and an adaptive procedure ensuring that the delay range was evenly sampled, while also being very sensitive to the individual indifference point. The use of the Kullback-Leibler divergence as a measure of change in the posterior distribution of discount parameter from before to after the learning block about another agent is elegant and fits with the fully Bayesian modeling and hierarchical estimation approach.

There are a several major issues that need to be addressed in a revised version of the manuscript.

Response: We greatly appreciate your feedback on our manuscript. Thank you for taking the time to review our work and for providing insightful comments that have enhanced our manuscript.

R2.1a: First, at a conceptual level, I think this study more about Theory of Mind (ToM) than direct social influence. The participants do not observe how others make delay discounting decision. Rather, they only get feedback if the decision that they made of another agent was correct or not. Thus, the participants are really building up a mental model of the agent that is being updated by feedback, which is an essential ToM capacity.

Response: Thank you for highlighting the possible role of theory of mind in social influence. The paradigm we used was designed to measure social influence (Garvert et al., 2015; Moutoussis et al., 2016; Reiter et al., 2021; Thomas et al., 2022), and conceptualised by previous authors as probing social influence on behaviour rather than theory of mind (ToM). We certainly acknowledge the potential significance of ToM in social influence in everyday life, yet it is worth noting that we did not manipulate levels of ToM ability in our study and therefore cannot draw direct conclusions about its role.

In addition, we found people displayed different susceptibility to impulsive and patient social influence. If ToM is indeed involved, it should theoretically be engaged in both scenarios, and therefore cannot solely explain the differences we observed in our findings.

However, we do not disregard the potential role of ToM in the process of social influence. For instance, in our new correlation analysis using scores from the factor analysis, we discovered that autistic and alexithymic traits were specifically correlated with susceptibility to patient

social influence rather than impulsive social influence, in older adults. This suggests that ToM might be partially involved in the process of social influence, with varying effects depending on the nature of the influence. Moreover, whilst studies of the ability for ToM in older adulthood are somewhat mixed with evidence of reduced or preserved ToM, no study, to our knowledge, has suggested enhanced ToM in older adulthood and here we show relatively greater susceptibility to impulsive influence than in younger adults. For these reasons, we suggest that ToM was not a generalisable driver of social influence for our study, although future studies that manipulate ToM levels would be needed to validate this hypothesis. We have now elaborated on the potential role of ToM in social influence in our discussion.

Discussion:

Future studies could also examine how much insight older adults have into their greater influence by impulsive others to understand whether and how such effects can be modified. There is also increasing empirical interest in the relationship between metacognition and mentalising, with possible computational and neural overlap between them⁶⁷⁻⁶⁹. In our task, one interpretation may be that being influenced by others' beliefs is linked to theory of mind ability. However, we found people displayed different susceptibility to impulsive and patient social influence. Even if theory of mind is indeed involved, it should theoretically be engaged in both scenarios. In addition, studies of older adults often suggest reduced or preserved theory of mind⁷⁰, whereas we found relatively enhanced susceptibility to social influence in older adults. Future studies could dissect and dissociate how and why metacognition, mentalising, and social influence drive any differences between older and young adults.

R2.1b: Furthermore, the inferred decisions of the agent do not affect the outcome of the participant - in fact they never really see the outcome of the agent. Because of the lack of joint decisions constructing and updating a mental model of the agent has no real consequences for the participant. From this perspective, it seems actually more surprising that participant shift their posterior of the discount rate toward that of the agent given that it has no consequence. Could there be a simpler explanation for this shifting effect? Maybe the participants are simply repeating/imitating the discounting choices from the previous learning block. Given that the learning is inconsequential for the participant, do we really know, that they really shifted their discounting parameter? A caveat that the authors might want to pick up in a revised Discussion.

Response: Thank you for raising this perspective. We agree it is surprising that participants shift even when these decisions may have negative impacts on their bonus payment. Because of this negative impact we believe it would be unlikely that they simply repeated what they had learnt in the self blocks. Neurally, previous work has suggested that a basic mechanism of plasticity in medial prefrontal cortex drives susceptibility to social influence, and it would be interesting for causal approaches to further probe this effect and see if there are separate brain areas implicated in patient and impulsive social influence. We agree it is helpful to consider different explanations for the effect and have added additional caveats to our discussion:

Discussion:

It is somewhat surprising that people shift their preferences to align with someone else when the shift could negatively impact their bonus payment. Social influence has been previously shown to operate in several different domains, including risk, and across different model-based and model-free analytical techniques^{13-16, 47-49}. An important next question is not how but why people shift., various explanations remain plausible. Participants might seek to learn about social norms¹² or gather information to reduce uncertainty¹⁴. These processes could be underpinned by more fundamental neural mechanisms, such as plasticity in the medial prefrontal cortex¹³.

R2.2: Rethinking this study in terms of ToM also unveils are shortcoming of the computational model. Simply fitting the same model to different block of trials and then comparing them, does not specify the mechanism of how the mental model interacts and influences in the self choices of the participant. Shall I think of the other agent's discounting parameter as a prior that is linearly weighted with the self discounting parameter? Or is it more like an initial prior for the self discounting parameter, whose influence dissipates over time? Or something entirely different? I would recommend that the authors develop some ideas about the cognitive mechanism of how the mental model of the other agent can affect self choices and then think about how this can be implemented in a computational model.

Response: Thank you for your comments on our computational modelling. The goal of our computational modelling is to precisely capture participants' discounting preferences in each Self experimental block, allowing us to accurately quantify changes in their preferences. This enables us to examine the impact of social influence with precision. The winning model is well-established from multiple other studies using similar paradigms (Moutoussis et al., 2016; Reiter et al., 2021), shows strong parameter recovery, and performs well in posterior predictive checks (see response to R2.3 below). As we did not focus on the role of mental models in social influence in this study, we are hesitant to make strong assumptions about them and to include them in our computational modelling.

It is also important to note that we used uninformative priors to estimate participants' discounting preferences in each Self block. This approach helps mitigate strong bias from their observations of others on the estimation of their own discounting preferences. We have updated our methods sections to highlight the use of uninformative priors:

Methods:

Model fitting

To ensure a more conservative estimation of all free parameters, the priors were reset at the beginning of each experimental block (i.e., the uninformative priors were used).

R2.3: Second, at the analysis level, the study is lacking a posterior predictive check (PPC) of the winning model. the model comparison shows decisive evidence for the KU model over the three alternatives, but can it really replicate the observed experimental data? And with that I not only mean a successful simulation of the self delay discounting, but also a successful mental simulation of the other agent and the shifting of the posterior of

discounting parameter afterwards. In order to achieve such a PPC a model-free behavioral index is needed as a target for the posterior simulations - something that the current version of the ms is also lacking. Specifically, one could look at the fixed option trials before and after the of the blocks and determined which of those changed. These fixed option trials could then be presented during the PPC to see if the model also changed its response in a similar way.

Response: Thank you for your suggestion to perform posterior predictive checks of the winning model. We have now used posterior predictive checks to evaluate the extent to which the posterior estimation of our winning model accurately replicates the key features of participant behaviour (e.g., their performances of learning about others' preferences). For this purpose, we used a post-hoc absolute-fit approach (Zhang et al., 2020), which factored in participants' actual decisions and option pairs, to generate predictions using the entire set of posterior Markov chain Monte Carlo (MCMC) samples of the winning model. To be specific, we let the winning model generate synthetic decisions repeatedly, matching the number of MCMC samples (i.e., 8000 times) for each trial and each participant, with the individual-level posterior parameters obtained through model estimation. Then, we analysed the synthetic data using the same methods as those applied to the actual data, using a linear mixed-effects model (LMM). This LMM incorporated fixed effects of age group (older vs young), other's preference (patient vs impulsive), and their interaction, along with a random subject-level intercept.

We found that the posterior prediction accurately replicated the key patterns in our behavioural data (**Figure S1**): learning the preferences of patient others was more accurate compared to impulsive ones (significant main effect of other's preference: $b = 0.03$, 95% CI = [0.02 0.05], $Z = 4.35$, $P < 0.001$), whereas there was no significant difference in learning performance between the two age groups (main effect of age group: $b = -0.01$, 95% CI = [-0.03 0.01], $Z = -1.04$, $P = 0.30$, $BF_{01} = 2.47$; age group x other's preference interaction: $b = -0.002$, 95% CI = [-0.02 0.02], $Z = -0.21$, $P = 0.83$, $BF_{01} = 6.21$). This indicates that our winning model could well capture our empirical findings, and we have added these new results to the manuscript.

In addition, we also used another model-free index, the choice proportions of delayed options, to gauge participants' temporal preferences and to examine whether we can replicate preference-shifting patterns observed using our winning model. We calculated the choice proportions of delayed options in each Self block (baseline, after learning the Impulsive Other, and after learning the Patient Other) and determined the differences between the baseline block and the post-learning blocks. Higher proportions of choosing delayed options indicate more patient temporal preferences. Positive values of the differences in choice proportions of delayed options indicate participants became more patient after learning others' preferences, while negative values indicate participants became more impulsive. To facilitate comparison with the original results, we flipped the data points after learning the Impulsive Other. This means that positive values in the differences in choice proportions of delayed options always indicate participants' preferences became more similar to those of the others. We analysed the differences in choice proportions of delayed options using the same method we applied to signed KL divergence (i.e., the model-based proxy of susceptibility to social influence). Specifically, we used a linear mixed-effects model (LMM) that included fixed effects of age group (older vs young), other's preference (patient vs impulsive), and their interaction, along with a random subject-level intercept.

We replicated all our main results (full results of LMM see **Table S5**). The LMM of differences in choice proportions of delayed options revealed a significant interaction between age group and other's preference ($b = -7.70$, 95% CI = [-11.91 -3.42], $Z = -3.54$, $P < 0.001$, **Figure S4**). Consistent with our previous findings, impulsive social influence had a greater impact on older adults compared to young adults (independent Wilcoxon signed-rank test: $W = 1823$, $Z = -2.83$, $r_{(140)} = 0.24$ [0.08 0.40], $P = 0.005$). Conversely, both young and older adults demonstrated similar levels of susceptibility to patient social influence ($W = 2866$, $Z = -1.74$, $r_{(138)} = 0.15$ [0.01 0.31], $P = 0.082$, $BF_{01} = 1.22$).

Additionally, although both young and older adults showed better learning regarding the preferences of patient others, they displayed distinct patterns in their susceptibility to social influence. Older adults were equivalently affected by both impulsive and patient social influence (paired Wilcoxon signed-rank test: $V = 1036$, $Z = -0.65$, $r_{(62)} = 0.07$ [0.00 0.33], $P = 0.518$, $BF_{01} = 5.89$). In contrast, young adults were more susceptible to patient than impulsive social influence ($V = 595$, $Z = -2.52$, $r_{(62)} = 0.31$ [0.09 0.53], $P = 0.012$). In summary, the model-free index of temporal preferences and susceptibility to social influence yielded results consistent with those obtained from the model-based measures.

New Figures and tables:

Figure S1. Posterior predictive checks of the winning KU model. Comparison of simulated learning accuracy shows that an equivalent learning performance of the others' preferences between the two age groups (no main effect of age group: $b = -0.01$, 95% CI = [-0.03 0.01], $Z = -1.04$, $P = 0.30$, $BF_{01} = 2.47$). In addition, both simulated young and older participants demonstrated better learning of the preferences of patient others compared to impulsive ones

(significant main effect of other's preference: $b = 0.03$, 95% CI = [0.02 0.05], $Z = 4.35$, $P < 0.001$). Large circles with border lines indicate the mean, error bars represent the standard error of the mean, dots show raw data, and asterisks denote the significant main effect of other's preference based on the linear mixed-effects model. Note that the vertical axis begins at 50%, representing the chance level. *** $P < 0.001$; ns: not significant. Red dots are the means of actual data.

Figure S4. Susceptibility to social influence quantified by model-free index, choice proportions of larger-and-later (LL) options. The model-free index of temporal preferences and susceptibility to social influence produced (i.e., choice proportions of delayed options) results that were consistent with the model-based measures (i.e., signed KL divergence). Compared to young adults, older adults were more susceptible to impulsive social influence ($W = 1823$, $Z = -2.83$, $r_{(140)} = 0.24$ [0.08 0.40], $P = 0.005$). Conversely, older and young adults were equivalently influenced by patient others ($W = 2866$, $Z = -1.74$, $r_{(138)} = 0.15$ [0.01 0.31], $P = 0.082$, $BF_{01} = 1.22$). Bars show group means, error bars are standard errors of the mean, dots are raw data, and asterisks represent significant two-sided between-group and within-group nonparametric t tests. * $P < 0.05$; ** $P < 0.01$; ns: not significant.

Table S5. LMM predicting unsigned differences in choice proportions of later-and-larger (LL) options.

Fixed effect	beta	95% CI	Z	P
(Intercept)	-0.47	[-2.63 1.69]	-0.43	0.67
Group (older vs young)	4.70	[1.70 7.68]	3.07	0.002
Others (patient vs impulsive)	5.80	[2.79 8.82]	3.78	<0.001

Group x Others	-7.70	[-11.91 -3.42]	-3.54	<0.001
----------------	-------	----------------	-------	--------

Note. LMM: linear mixed-effects model; 95% CI: 95% confidence interval.

R2.4: Similarly, it would good to verify that the modeling the choices of the participant for the other agents results in discounting and softmax parameters that a similar to the ones that generated the choice behavior of the other agent. This would be another step to verify that the learning process not only matches in terms of accuracies, but also in terms of the hidden variables that generated the simulated behavior.

Response: Thank you for your suggestion to examine the similarities between parameters estimated from the winning model and those used to generate simulated choices. To verify whether our winning model can accurately capture the learning process, we conducted a correlation analysis between the means of discounting distribution () estimated from the Bayesian winning model and the hyperbolic discount rates () derived from the non-Bayesian preference-temperature model used for simulating others' choices. We found that these two parameters were highly correlated in both conditions (**Figure S2**), whether the person was more impulsive ($r_{S(152)} = 0.91$ [0.88 0.94], $P < 0.001$) or more patient ($r_{S(152)} = 0.82$ [0.76 0.86], $P < 0.001$). This suggests that our winning model can effectively reflect the underlying learning process.

Figure S2. Correlations between parameters estimated from the winning model and those used to simulate others' choices. To assess our winning model's accuracy in capturing the learning process, we analysed the correlations between the means of discounting distribution (km) from the Bayesian winning model fitted with experimental data and the hyperbolic discount rates (k) from the non-Bayesian preference-temperature model that was used to simulate other's choices as task stimuli. The results showed a significant correlation in both conditions, for both impulsive ($r_{S(152)} = 0.91$ [0.88 0.94], $P < 0.001$) and patient ($r_{S(152)} = 0.82$ [0.76 0.86], $P < 0.001$) others. This suggests that our model accurately reflects the learning process.

R2.5: Finally, in the Result section information the group mean parameters for the KU model for all blocks are missing.

Response: Thank you for raising this. We have updated the relevant paragraph as follows and included a table in the supplement:

Results:

Baseline impulsivity does not differ with age

We found no difference in either mean (young group mean [SE] = -4.79 [0.22], older group mean [SE] = -5.16 [0.25]; [...] or standard deviation (young group mean [SE] = 1.37 [0.06], older group mean [SE] = 1.47 [0.06]; [...]) of the discounting distribution at the baseline between age groups.

Supplementary information:

Table S3. Model parameters from the winning model for each experimental block.

	Young		Older	
	km	ku	km	ku
Self baseline	-4.79[0.22]	1.37[0.06]	-5.16[0.25]	1.47[0.06]
Impulsive Other	-3.34[0.25]	1.77[0.06]	-3.65[0.21]	1.63[0.04]
Self after Impulsive Other	-4.87[0.26]	1.31[0.06]	-4.76[0.24]	1.18[0.04]
Patient Other	-6.58[0.17]	1.93[0.07]	-6.19[0.15]	1.65[0.03]
Self after Patient Other	-5.48[0.22]	1.37[0.07]	-5.49[0.21]	1.18[0.04]

Note. km: mean of discounting distribution; ku: standard deviation of discounting distribution. The numbers shown are group means [standard errors].

R2.6: Third, I agree with the authors that a visualization of this shift in posterior distributions is needed for an understanding, but the way in which it is shown in Figure 2C is utterly confusing. This figures needs to be redesigned from scratch. For instance, I think this figure conflates different posteriors over discount rates with the resulting DKL and labeling the DKLs as “impulsive” and “patient” just as other two agents does not really help. A label of “positive” or “negative” would be less confusing. The fact that the direction of the shift in posteriors (in the upper panel of 2C) is away from the posterior of the other agent is rather unintuitive and needs to be explained much better (one would expect that the posterior moves toward the posterior of the other as a expression of social influence).

Response: Thank you for your comments on our figures and we apologise for any lack of clarity and appreciate your suggestions. The label ‘impulsive_{DKL}’ indicates the changes in discounting distributions after learning the preference of Impulsive Other, while ‘patient_{DKL}’ signifies the changes after learning the preference of Patient Other. These values could be positive (i.e., participants’ preferences became more similar to those of others) or negative (i.e., participants’ preferences became more dissimilar to those of others). Therefore, the labels ‘positive_{DKL}’ and ‘negative_{DKL}’ might be less appropriate here.

The upper panel of Fig. 2C in the original manuscript illustrates the scenario where participants’ preferences diverge from those of others following social influence. We used negative D_{KL} values to indicate when participants’ preferences became more dissimilar to those of others

(see response to R2.7), which means people were **not susceptible** to social influence. Conversely, positive $D_{\#}$ values signify when participants' preferences became more similar to those of others, which means people were susceptible to social influence. This approach allows us to use a single quantitative measure to capture both scenarios, regardless of whether people are susceptible to social influence or not.

We have updated the figure as follows for better illustration:

Figure 2. Susceptibility to social influence quantified by the signed Kullback-Leibler divergence (D_{KL}). ... (C) Example of shifts in self discounting distributions after learning about the preferences of an Impulsive Other and a Patient Other. (upper) In this example, the participant firstly completed a baseline block to assess their own baseline temporal preference (Self1, dark green solid line) before learning about the preference of an Impulsive Other (Other1, blue dashed line). After learning the preference of Impulsive Other, they completed another block making their own intertemporal choices (Self2, green solid line). For this participant,

their preference shifted away from that of Impulsive Other ('Impulsive D_i '), meaning that the participant's own temporal preference became less similar to that of the Impulsive Other (represented by a negative D_i value). (lower) Following this, the participant learnt about the preference of a Patient Other (Other2, yellow dashed line) before making their own intertemporal choice again (Self3, light green solid line). For this participant, their preference shifted towards that of Patient Other ('Patient D_i '). The positive D_i value here means that the participant's preference became more similar to that of the Patient Other after observing their preference.

R2.7: The "normalization" step of DKL (which actually just flips the sign of DKL, so normalization may not be the appropriate term here after all) needs to be explained much better, possibly with an example, e.g. "if the difference between other and self k_m is positive, and the difference between before and after self k_m is negative, then DKL is negative, which means ..." Also, because DKL and the shift in posterior mean is not the same scale, I think it makes more sense not to include both in Figure 2C, but rather split them up into two figures.

Response: Thank you for your feedback on our explanation of measures between posterior probability distributions. We acknowledge that the term 'normalised' may not be the most appropriate term for the process of flipping the sign of $D_{i\#}$. Therefore, we have opted to use the term 'signed $D_{i\#}$ ' throughout the manuscript for greater clarity.

We have added a more detailed description of the process of flipping the sign of D_i :

Methods:

Signed Kullback-Leibler divergence

... Positive D_i values signify a shift in participants' discounting preferences towards those of the other agents, while negative D_i values indicate a shift away from them, compared to the baseline discounting preferences (see **Figure 2C**):

(Formula 8)

where k_m represents the mean of discounting distribution estimated using the KU model, and the subscript i denotes the number of Other blocks (i.e., 2 or 4). For example, if a participant's discounting preference shifts to be more negative (i.e., more patient) after exposure to the discounting preference of a patient other agent, this would be reflected by a positive D_i value. More specifically, if the difference between the other and self baseline k_m is negative, and the difference between self after observation and self baseline k_m is also negative (i.e., when the differences are of the same sign), then D_i is positive, which means that the participant becomes more similar to others. Conversely, negative D_i values signal a divergence in the participants' discounting preferences from those of the other agents. For example, if the difference between the other and self baseline k_m is positive, while the difference between self after observation and self baseline k_m is negative (i.e., when the differences are of opposite signs), then D_i is negative, which indicates that the participant becomes more dissimilar to others.

R2.8: Fourth, Why were these particular 4 models included in the model comparison and not the “classical set” of discounting models that prevail in the literature)? (summarized and tested e.g. in these papers Doyle & Chen, 2012, The Wages and Waiting of Simple Models of Delay Discounting, Econometric Modeling; McKerchar et al., 2009, A comparison of four models of delay discounting in humans, Behav Processes) Of course, these are the ones from the Moutoussis paper that inspired the study, but a more detailed explanation of these alternative models and their rationale should be included. Overall, I think that the reference to the Moutoussis paper as a template for the study and the modeling efforts should be made more explicit.

Response: Thank you for your feedback on our model space. We indeed followed the reasoning from the Moutoussis paper and showed good parameter recovery and PCC from the winning model. We have now made the link between our paper and the Moutoussis paper more explicit.

Methods:

We considered a set of models applicable to our paradigm based on previous work¹⁴ examining social influence to constrain our model space and for the results to be comparable across studies.

Minor point:

R2.9: There is a reference missing on page 26 (lines 542) where the authors talk about the adaptive method of generating option choices and estimating the indifference point efficiently.

Response: Thank you for spotting this, and the relevant reference has been added:

Methods:

Optimisation of choice pairs

... Previous studies have demonstrated that this method is capable of generating

more reliable estimates of the value while requiring fewer trials^{74, 75}.

R2.10: Overall, I think the writing in the Results and Methods section is quite dense, especially for readers that are not well-verse in Bayesian analysis and how it should be used to analyze and model data. While I don't have an particulate section in mind, I think the text would benefit from a few more explanatory sentences about the analyses, what they can achieve, and what they are good at. Short statements could go in the Results, but a bi more longer explanation would help to understand the Methods better.

Response: Thank you for your feedback. We have fully revised these sections to include more explanatory sentences guiding readers through the analysis.

Reviewer #3 (Remarks to the Author):

MS# NCOMMS-23-55716

MS Title: Older adults are more susceptible to impulsive social influence

Summary: This manuscript describes a single study of temporal discounting with a social manipulation in younger (aged 18-31) and older (aged 60-80) adults. Participants completed 5, 50-trial blocks of a temporal discounting task. During the first, third, and fifth blocks, participants made choices for themselves and received no feedback about those choices (Self blocks). During the second and fourth blocks, participants made choices for another person and received feedback about whether that choice was correct or not (Other blocks). They examined whether learning about the choices of others (during Other blocks) would influence the choices they made for themselves (Self 2 and Self 3 blocks). They found there were no age differences in initial discount rates (Self 1 block), nor for learning during the Other blocks. When they examined the influence of others, they found that both age groups were influenced by the patient partner, but only the older adults were influenced by the impatient partner.

Critique: Thank you for the opportunity to review this manuscript. This is a very interesting study of the impact of social influence on temporal discounting using sophisticated modeling techniques. As noted by the manuscript, there is a dearth of data examining social influence in older adults, and thus this manuscript is poised to make an important contribution to the field. However, there are several concerns about the data analysis and how it is presented that should be addressed.

Response: We sincerely appreciate your feedback on our manuscript. Thank you for taking the time to review our work and for offering insightful comments that have significantly improved our manuscript.

Major concerns:

R3.1a: My first major concern is that some of terminology used in the manuscript is somewhat misleading or confusing. First, social discounting is term that has been used to describe how people treat social distance – see Jones & Rachlin, 2006 reference below for an example of this. The phenomena studied in this manuscript is temporal discounting, not social discounting, thus it is confusing to call this a “social discounting task”.

Response: Thank you for pointing out this important terminology issue. We acknowledge the terminology to describe social influence as social discounting could be confusing for readers. We have now changed the task name to ‘delegated intertemporal choice task’, which is more informative and consistent with previous work (Garvert et al., 2015; Moutoussis et al., 2016; Nicolle et al., 2012). We have made updates throughout the manuscript when the task is described, for example:

Methods:

Delegated inter-temporal choice task

Participants completed a *delegated inter-temporal choice task* where they learnt about impulsive and patient others after completing their own temporal discounting preferences (see **Figure 1A**). 1...]

Delegated inter-temporal choice task-specific questionnaires

Participants were asked four questions regarding their confidence in learning the other two agents' preferences, as well as their perceived similarity to these agents. 1...]

R3.1b: Second, using the term “impulsive” to describe people who are not willing to wait for LL rewards is imprecise. Impulsivity is characterized by a lack self-control – someone can have self-control and still choose not to wait for the LL reward. The term “impatient” may help avoid this by focusing on the willingness to wait, which is what is measured by temporal discounting tasks.

Response: Thank you for highlighting potential concerns with the use of the term ‘impulsive’. We are aware of divergent interpretations within the field regarding the concept of impulsivity. However, there is a substantial body of research employing temporal/delay discounting tasks to gauge people’s preferences for larger-and-later rewards, and to consider them as manifestations of temporal impulsivity (Peters & Büchel, 2011; Rung & Madden, 2018). Therefore, we have chosen to retain this term. However, to mitigate potential confusion, we have now provided a more explicit definition of impulsivity in our manuscript:

Introduction:

Humans vastly differ in how impulsive or patient they are (i.e., their willingness to wait for larger rewards in the future). 1...]

R3.1c: Finally, the terms “emotional motivation,” “emotional apathy,” and “emotional sensitivity” are used interchangeably to describe the “emotional sensitivity” subscale of the Apathy motivation index. It would be helpful to use one term consistently throughout the manuscript – and it seems like emotional sensitivity is the most accurate term.

Response: We agree that using the consistent term would aid readers in better understanding the manuscript. We have now adopted the term ‘**emotional sensitivity**’ wherever referring to this specific subscale of the Apathy-Motivation Index as this is the terminology used by the questionnaire designers. In all other places in the manuscript, we use the term ‘emotional motivation’ to more accurately describe the process of emotional motivation.

R3.2: My second major concern is with the way that the results are presented in the title and abstract. Based on reading these, one would think that only older adults are influenced by others, and that this only occurs for impulsive others. However, the results presented show that both age groups are influenced by patient others, and only older adults are influenced by impatient others. So, the “norm” here is to be influenced by others, and the deviation is actually the younger adults who are less likely to be influenced by impatient others. This should be clear in the abstract and title; otherwise it is misleading.

Response: Thank you for this feedback on our presentation of the results. We agree that our manuscript could be edited to describe our results without any ambiguity. We believe it is important to highlight the relative difference between young and older adults in impulsive social influence as a central result for several reasons. First, in our design susceptibility to social influence is continuous so the difference between young and older adults, who we carefully matched, is relative. Second, we used two distinct groups to maximise power. Comparing these two groups shows a relative difference in impulsive social influence. Third, susceptibility to impulsive and patient social influence may operate through separate mechanisms and therefore the comparison on impulsive influence between young and older adults is valid. In a new analysis (also in response to **R3.5**) we found that individual differences in susceptibility to impulsive social influence were correlated with traits related to affective empathy and emotional motivation. Conversely, individual variations in susceptibility to patient social influence correlated with autistic and alexithymic traits.

We believe that emphasising the relative difference between young and older adults could be acknowledged more clearly in the key conclusions of our manuscript. We have therefore now revised our title to '**Older adults are relatively more susceptible to impulsive social influence than young adults**'. In addition, the relevant parts in the abstract have also been updated.

Abstract:

We used the signed Kullback-Leibler divergence to quantify the magnitude and direction of social influence. We found that, compared to young adults, older adults were relatively more susceptible to impulsive social influence.

Discussion:

Another key finding was that younger adults did not show susceptibility to impulsive social influence, it neither made them more impulsive or more patient, instead they, on average, did not shift their preference. There are several explanations for this finding which could be probed in future research. For example, some studies have found that compared to adolescents, young adults are less susceptible to social influence across several domains (reviewed in⁵⁰). Reduced social influence in young adults has been attributed to their reduced normative pressure to conform to others or their greater certainty about their own preferences, making them less likely to be influenced than adolescents¹⁶. It is interesting here that both young and older adults were susceptible to patient social influence, and that older adults who anecdotally may be expected to hold stronger belief certainty, were influenced in both domains. Again, together these studies highlight the importance of future work considering the whole lifespan.

R3.3: My third major concern is with the main result reported. Careful examination of Figure 2a suggests that there is an outlier in the young adult impulsive group – one person who showed a dramatic shift away from the impatient person. Does the significant difference hold if this outlier is removed? Were any diagnostics conducted to ensure that this (and other instances) did not overly influence the analyses?

Response: Thank you for your suggestion to examine the potential influence of outliers on our results. Our original analyses sought to be fully inclusive and not omit any data points as

those representing potential outliers would still be valid behaviour in our task. However, we agree it is important to represent additional robustness by checking the influence of extreme data points on the main findings. After removing outliers, defined as any data points falling below or above three standard deviations from the group mean, we re-ran our analyses, and all of our main results remained unchanged. Older adults were still relatively more influenced by impulsive social influence compared to young adults, and no difference was observed in terms of susceptibility to patient social influence between young and older adults. We added these new analyses to the supplement:

Supplementary results:

*We conducted an additional control analysis to confirm our results were robust to the influence of any extreme data point. We re-ran all of our main analyses excluding any data points that fell below or above three standard deviations from the group mean. All previously reported results were replicated (see **Tables S6 and S7**, and **Figure S5**).*

*In addition, all previously reported correlations between participants' self-reported socio-affective traits and their susceptibility to social influence remained (see **Table S9**).*

Figure S5. Susceptibility to social influence quantified by the signed KL divergence () after removing outliers. After removing outliers (i.e., any data points falling below or above three standard deviations from the group mean, all our main results stayed the same. Older adults were more susceptible to impulsive social influence than young adults ($W = 1861$, $Z = -2.44$, $r(138) = 0.21$ [0.05 0.36], $P = 0.015$). Conversely, both older and young adults exhibited comparable susceptibility to patient social influence ($W = 2655$, $Z = -1.17$, $r(136) =$

0.10 [0.00 0.27], $P = 0.24$, $BF_{01} = 3.20$). Bars show group means, error bars are standard errors of the mean, dots are raw data, and asterisks represent significant two-sided between-group and within-group nonparametric t tests. * $P < 0.05$; *** $P < 0.001$; ns: not significant.

Table S6. LMM predicting susceptibility to social influence, after removing outliers

Fixed effect	beta	95% CI	Z	P
(Intercept)	0.01	[-0.14 0.16]	0.12	0.90
Group (older vs young)	0.33	[0.12 0.55]	3.09	0.002
Others (patient vs impulsive)	0.45	[0.24 0.66]	4.15	<0.001
Group x Others	-0.42	[-0.72 -0.12]	-2.76	0.006

Note. LMM: linear mixed-effects model; 95% CI: 95% confidence interval.

Table S7. LMM predicting susceptibility to social influence, with self baseline temporal impulsivity as covariates (centred around the grand mean), after removing outliers

Fixed effect	beta	95% CI	Z	P
(Intercept)	0.04	[-0.15 0.16]	0.06	0.96
Group (older vs young)	0.31	[0.10 0.53]	2.91	0.004
Others (patient vs impulsive)	0.44	[0.23 0.66]	4.01	<0.001
Self baseline km	0.01	[-0.07 0.10]	0.32	0.75
Group x Others	-0.40	[-0.71 -0.09]	-2.57	0.01
Group x Self baseline km	-0.11	[-0.23 0.00]	-2.02	0.04
Others x Self baseline km	0.02	[-0.11 0.15]	0.24	0.81
Group x Others x Self baseline km	0.10	[-0.01 0.29]	1.13	0.26

Note. LMM: linear mixed-effects model; 95% CI: 95% confidence interval; km: the estimated mean of discounting distribution from the winning model.

Table S9. Correlations between the factors and signed KL divergence (D_{KL}), after excluding outliers

			Autistic & alexithymic traits	Psychopathic traits	Affective empathy & emotional motivation
Young	Impulsive	$r_{S(65)}$	0.09 [-0.15 0.33]	0.02 [-0.22 0.26]	-0.13 [-0.36 0.11]
	D_{KL}	BF_{01}	5.52	7.17	4.57
	Patient	$r_{S(68)}$	-0.03 [-0.26 0.21]	-0.07 [-0.30 0.17]	-0.09 [-0.31 0.15]
	D_{KL}	BF_{01}	7.46	6.37	6.22
Older	Impulsive	$r_{S(70)}$	-0.08 [-0.31 0.15]	0.05 [-0.18 0.28]	0.28 (0.05 0.48)*
	D_{KL}	BF_{01}	6.06	5.29	0.34
	Patient	$r_{S(65)}$	0.35 (0.12 0.55)**	0.16 [-0.09 0.38]	-0.11 [0.34 0.13]
	D_{KL}	BF_{01}	0.17	1.98	2.95

Note. $r_{S(df)}$: Spearman's Rho correlation coefficients with degrees of freedom; 95% confidence intervals are indicated in square brackets. BF_{01} indicates the strength of evidence for the null hypothesis. * $p < 0.05$, ** $p < 0.01$

R3.4: Finally, how does this relate to the recent publication showing that susceptibility to social influence declines with age? See Castellon et al., 2023 below. Since this is a new

area of research, it would be helpful to discuss why these two studies suggest different patterns of susceptibility to social influence.

Response: Thank you for highlighting other important work related to this interesting research topic. Despite this related work, we believe our study still carries significant novelty in multiple domains.

Castrellon and colleagues used experience sampling surveys to examine the effect of the presence of others on self-control in three distinct age groups. They showed that compared to young adults (aged 18-30 years), middle-aged (aged 31-59 years) and older adults (aged 60-80 years) were more likely to practise self-control when there were others present enacting the desire. However, there are several critical differences between our study and theirs, supporting empirical novelty in our results. Firstly, the conceptualisation of socially conforming to others differs between the two studies. In the study by Castrellon et al. (2024), social conformity was defined as the inability to resist desire when there were others present enacting the same desire, emphasising external behaviours rather than underlying preferences. On the other hand, our study tests how people's underlying preferences are influenced by observing others' economic preferences, and social conformity was defined as participants' preferences shifted towards those of others. Given the different pattern of results in the two studies, our findings highlight the importance of different dimensions of social conformity and perhaps that people might change control over their actions without changing their preferences.

Secondly, the study design is distinctive between the two manuscripts. Castrellon et al. (2024) took advantage of the experience-sampling technique to achieve external validity. However, this approach involved a trade-off as the heterogeneity of scenarios—such as varying types of desires, numbers of others present, and types of social interactions—was not controlled. We adopted a laboratory design that, while potentially limiting external validity, enabled us to apply precise and advanced computational modelling. This approach helped us reveal the latent factors driving susceptibility to patient and impulsive influence.

Finally, the composition of participants differs. In Castrellon et al.'s study (2024), across the three samples included in the analysis, there were 13 older adults (aged 60-80 years) out of total 157 participants (8.3%), whereas we had 78 older adults (aged 60-80 years) out of total 154 participants (50.6%). The larger sample of older adults aids us in drawing stronger conclusions about young and older adults as distinct groups, although we agree future work would benefit from also incorporating a longitudinal or large lifespan approach. We have now added this point as a future direction and limitation in our discussion.

Regarding Bixter and Rogers (2019) we already briefly discussed their paper in our original manuscript, but we inadvertently omitted the reference during editing. We have now added the reference and again highlighted why our study provides novel contributions. In particular, in Bixter and Rogers participants choices were not incentivized directly and differed between young and older adults (young adults received course credit and older adults received a fixed \$30 regardless of their choices). Consequently, their choices may not reflect their true preferences⁴⁹. Secondly, because we independently measured patient and impulsive social influence through multiple trials we were able to fit detailed computational models that can be generalised to different contexts and populations and can be more sensitive when uncovering

latent preferences. Finally, Bixter and colleagues asked participants to discuss their preferences in matched dyads that were video recorded. We show that simply learning about another's preference differently impacts young and older adults' susceptibility to social influence. We again believe these important differences highlight strengths and weakness of both approaches in this emerging research field.

Discussion:

However, another study using a collaborative delay discounting task observed no discernible difference in susceptibility between the two age groups⁴⁴. Notably, in this latter study, participants' choices were not incentivized and were unmatched, as young adults received course credit and older adults received \$30 regardless of their decisions. Consequently, their choices may not have reflected true preferences⁴⁵. We were able to fit detailed computational models of incentivized choices, and separately measure susceptibility to patient and impulsive influence. Another recent study using experience sampling showed that compared to young adults (aged 18-30 years), middle-aged (aged 31-59 years) and older adults (aged 60-80 years) were more likely to practise self-control when others were present enacting the desire, suggesting older people were less susceptible to social influence than young adults. The distinct patterns between this study and ours may also be attributed to several factors: the differing composition of young and older adult participants, different aspects of social conformity investigated, and different experimental approaches. Together these studies highlight the importance of considering the multidimensional aspect of social influence and utilising difference techniques to understand how susceptibility to social influence evolves over the course of life. It will be important for future work to examine dynamic fluctuations in susceptibility to social influence across the lifespan. Here and in other studies researchers have often focussed on cross-sectional samples for feasibility and increased power, yet longitudinal studies and those that include mid-life samples are crucial for enhancing our understanding of the process of social influence.

Minor concerns:

R3.5: p. 15/Fig. 2B – Only the correlation between older adults change in discounting and emotional sensitivity is shown here. Were these correlations explored for other groups? It seems the the group that deviates from the norm is actually the younger adults – not the older adults. Do those younger adults who avoid the influence of a more impatient peer show differential scores on these measures?

Response: Thank you for your query. We did examine the relationships between emotional sensitivity and susceptibility to social influence in young adults, as reported in the original manuscript (p. 15-16, lines 280-288). However, we did not find any significant correlations, regardless of whether the influence was impulsive (Spearman: $r_{s(66)} = 0.09 [-0.15 \ 0.32]$, $P = 0.472$, $BF_{01} = 9.08$) or patient (Spearman: $r_{s(69)} = -0.01 [-0.25 \ 0.22]$, $P = 0.909$, $BF_{01} = 7.37$).

In this study we chose to focus on emotional motivation as theoretically, we hypothesised that this subscale may most accurately explain the variability in being influenced by others, as we highlight in the manuscript. We also were mindful to limit the possible number of multiple

comparisons to balance between type 1 and type 2 errors. In focussing on this subscale, we certainly did not want to be untransparent, as in most large studies we indeed collected additional measures but had no strong theoretical reason to focus on those instead.

To ensure we could include all measures, we have now taken a data-driven rather than theory-driven approach and conducted an exploratory factor analysis including all collected measures. This analysis identified a three-factor latent structure, with the third factor 'Affective empathy & emotional motivation', incorporating the emotional sensitivity subscale from the AMI (Apathy-Motivation Index) and the affective empathy subscale from the QCAE (Questionnaire of Cognitive and Affective Empathy). This finding supports our original hypothesis of a strong link between emotional motivation and affective empathy. Using scores from the 'Affective empathy & emotional motivation' factor also confirmed our earlier results: a significant relationship between traits related to emotional motivation and susceptibility to social influence was observed only in older people when the influence was impulsive. Furthermore, the direction of this correlation aligns with our previous finding: older adults with higher levels of emotional motivation (and affective empathy) were more susceptible to impulsive social influence. Amongst young adults, no significant correlations were found between scores from the 'Affective empathy & emotional motivation' factor and susceptibility to social influence, irrespective of whether the influence was impulsive or patient.

In addition, this analysis also revealed a new result: a significant positive correlation between the factor 'Autistic & alexithymic traits' scores and people's susceptibility to patient social influence amongst older participants, which suggests older people with higher levels of autistic and alexithymic traits are more influenced by patient social influence. This correlation was not found amongst young people or for impulsive social influence. We have added this new analysis to the results:

Results:

*Finally, we examined how individual variations in socio-affective traits modulated people's susceptibility to social influence. For this purpose, we performed an exploratory factor analysis on the self-report questionnaires completed by participants (see Methods). This enabled us to identify latent patterns of behaviour measured across the questionnaires (e.g., affective empathy and emotional sensitivity scores were highly correlated, $r_{(150)} = -0.65$), facilitating both conceptual interpretation and statistical inference by reducing the number of comparisons. The factor analysis uncovered three distinguishable dimensions across the subscales of the questionnaires included (**Figure S6A**). Factor 1 (Autistic & alexithymic traits) involved high loadings (> 0.40) from measures related to autism, alexithymia, cognitive empathy and social apathy, Factor 2 (Psychopathic traits) encompassed high loadings (> 0.40) from measures of psychopathic traits, and Factor 3 (Affective empathy & emotional motivation) included high loadings (> 0.40) from measures of affective empathy and emotional motivation. Comparing the two age groups on these factors showed that no overall age-related difference was observed in the factor 'Autistic & alexithymic traits' (young mean [SE] = -0.12 [0.11], older mean [SE] = 0.11 [0.10]; $W = 2501$, $Z = -1.42$, $r_{(150)} = 0.12$ [0.01 0.26], $P = 0.155$, $BF_{01} = 1.96$). However, older people scored significantly lower in both the factor 'Psychopathic traits' (young mean [SE] = 0.25 [0.11], older mean [SE] = -0.24 [0.10]; $W = 3944$, $Z = -3.89$, $r_{(150)} = 0.32$ [0.17 0.46], $P < 0.001$) and the factor 'Affective*

empathy & emotional motivation' (young mean [SE] = 0.25 [0.11], older mean [SE] = -0.24 [0.09]; $W = 3833$, $Z = -3.48$, $r_{(150)} = 0.28 [0.12\ 0.44]$, $P < 0.001$).

We correlated the scores from each factor with people's tendency to socially conform to others (**Figure S6B**). We found a significant positive correlation between impulsive $_{DKL}$ and the factor 'Affective empathy & emotional motivation' scores amongst older participants (Spearman: $r_{s(71)} = 0.29 [0.06\ 0.48]$, $P = 0.014$; FDR-corrected for three factor comparisons $P = 0.043$), but not amongst young people ($r_{s(66)} = -0.13 [-0.36\ 0.11]$, $P = 0.30$, $BF_{01} = 5.33$). Moreover, the association between the factor 'Affective empathy & emotional motivation' scores and impulsive social influence was significantly stronger in older adults than in young adults ($Z = 2.45$, $P = 0.014$). There was no statistically significant correlation found between patient $_{DKL}$ and the factor 'Affective empathy & emotional motivation' scores in either older ($r_{s(66)} = -0.11 [-0.34\ 0.13]$, $P = 0.36$, $BF_{01} = 3.25$) or young ($r_{s(69)} = -0.06 [-0.29\ 0.18]$, $P = 0.63$, $BF_{01} = 6.48$) participants. The findings collectively suggest a specific association between socio-affective traits and susceptibility to impulsive social influence among older adults. Older adults who are more susceptible to impulsive social influence also report being more affectively empathetic and emotionally motivated.

Additionally, the factor 'Autistic & alexithymic traits' showed a significant positive correlation with patient $_{DKL}$ amongst older adults ($r_{s(66)} = 0.34 [0.11\ 0.54]$, $P = 0.004$; FDR-corrected for three factor comparisons $P = 0.013$), but not amongst young adults ($r_{s(69)} = -0.04 [-0.27\ 0.20]$, $P = 0.76$, $BF_{01} = 7.15$). No significant association was observed between impulsive $_{DKL}$ and the scores from the 'Autistic & alexithymic traits' in either older ($r_{s(71)} = -0.07 [-0.30\ 0.16]$, $P = 0.54$, $BF_{01} = 6.59$) or young adults ($r_{s(66)} = 0.11 [-0.13\ 0.34]$, $P = 0.35$, $BF_{01} = 4.90$). This suggests that older people with higher levels of autistic and alexithymic traits are more likely to be affected by patient social influence. Notably, Bayesian analysis also showed that psychopathic traits did not account for individual variations in susceptibility to both impulsive and patient social influence, in both young and older adults (all $BF_{01}s > 5.20$; see **Table S8**). Moreover, all these results remained the same after removing outliers (**Table S9**).

Figure S6. Individual variations in susceptibility to social influence amongst older adults are related to socio-affective traits. (A) Factor loadings for each subscale of the questionnaires completed. Exploratory factor analysis uncovered three distinct dimensions related to social affective cognition and behaviour: autistic & alexithymic traits, psychopathic traits, and affective empathy & emotional motivation. Note: the loadings displayed here are absolute values. AQ: Autism Quotient; AMI: Apathy-Motivation Index; TAS: Toronto Alexithymia Scale; QCAE: Questionnaire of Cognitive and Affective Empathy; SRP: Self-Report Psychopathy scale. (B) The factors ‘Autistic & alexithymic traits’ and ‘Affective empathy & emotional motivation’ related to susceptibility to social influence in older adults, depending on the nature of social influence. Older people with higher levels of autistic and alexithymic traits are more susceptible to patient social influence, while those who are more affectively empathetic and emotionally motivated display greater susceptibility to impulsive social influence.

Table S8. Correlations between the factors and signed KL divergence (D_{KL})

			Autistic & alexithymic traits	Psychopathic traits	Affective empathy & emotional motivation
Young	Impulsive	$r_{S(66)}$	0.11 [-0.13 0.34]	0.01[-0.23 0.25]	-0.13 [-0.36 0.11]
	D_{KL}	BF_{01}	4.90	6.85	5.33
	Patient D_{KL}	$r_{S(69)}$	-0.04 [-0.27 0.20]	-0.07 [-0.30 0.17]	-0.06 [-0.29 0.18]
		BF_{01}	7.15	6.40	6.48
Older	Impulsive	$r_{S(71)}$	-0.07 [-0.30 0.16]	0.03[-0.20 0.26]	0.29 (0.06 0.48)*
	D_{KL}	BF_{01}	6.59	6.51	0.25
	Patient D_{KL}	$r_{S(66)}$	0.34 (0.11 0.54)**	0.11[-0.13 0.34]	-0.11 [0.34 0.13]
		BF_{01}	0.22	5.22	3.25

Note. $r_{S(df)}$: Spearman's Rho correlation coefficients with degrees of freedom; 95% confidence intervals are indicated in square brackets. BF_{01} indicates the strength of evidence for the null hypothesis. * $P < 0.05$, ** $P < 0.01$.

Methods:

Exploratory factor analysis

We performed an exploratory factor analysis on the questionnaire subscales using the 'fa' function (from the {psych} v2.4.3 package) in R v4.2.1. We incorporated all the subscales from the Autism Quotient (AQ), Apathy-Motivation Index (AMI), Toronto Alexithymia Scale (TAS), Self-Report Psychopathy scale (SRP), and the Questionnaire of Cognitive and Affective Empathy (QCAE). To extract factor loadings, we used maximum likelihood estimation with an oblimin rotation. Regarding the determination of the number of factors (using the 'fa.parallel' and 'vss' functions from the {psych} v2.4.3 package), the Kaiser rule (eigenvalue > 1) pointed toward a 2-factor solution, the very simple structure (VSS) complexity 2 criterion implicated a 3-factor solution, examination of the scree plot indicated a 3-factor solution, and parallel analysis suggested a 4-factor solution. After weighing parsimony and interpretability of the latent structure, we settled on the 3-factor solution. This 3-factor latent structure explained 50.55% of the variance in the measures, with moderate correlations with each other (highest $r = 0.25$). Individual scores for each factor were calculated using Thurstone's at the participant level. These scores were subsequently correlated with the signed KL divergence using Spearman's Rho correlation coefficients.

R3.6: Fig. 2C – The caption is difficult to follow. Perhaps partially because Self1, Self2, etc are not described. It might be helpful to refer to Fig 1a. Also, the top description ends with the phrase “resulting in a negative value” – a negative value of what? The bottom description similarly ends with the phrase “Resulting in a positive value”. I think this figure/caption is trying to communicate that this person shows a shift away from the impatient other (and the negative value is for DKL) and then shows a shift towards the patient other (and the positive value is for DLK), but this is not clear.

Response: We apologise for the lack of clarity in our caption presentation and thank you for your suggestion to improve it. We have edited the figure, also in response to Reviewer 2, and included a more detailed legend. We modified our legend to better correspond to the Fig. 1A. The top description intended to express ‘... resulting in a negative value of D_{KL} ’, meaning that this person’s preference shifted away from that of the Impulsive Other. Similarly, the bottom description intended to express ‘... resulting in a positive value of D_{KL} ’, indicating that this person’s preference shifted towards that of the Patient Other. The updated figure and caption are as follows:

Figure 2. Susceptibility to social influence quantified by the signed Kullback-Leibler divergence (D_{KL}). ... (C) Example of shifts in self discounting distributions after learning about the preferences of an Impulsive Other and a Patient Other. (upper) In this example, the participant firstly completed a baseline block to assess their own baseline temporal preference (Self1, dark green solid line) before learning about the preference of an Impulsive Other (Other1, blue dashed line). After learning the preference of Impulsive Other, they completed another block making their own intertemporal choices (Self2, green solid line). For this participant,

their preference shifted away from that of Impulsive Other ('Impulsive_{DKL}'), meaning that the participant's own temporal preference became less similar to that of the Impulsive Other (represented by a negative_{DKL} value). (lower) Following this, the participant learnt about the preference of a Patient Other (Other2, yellow dashed line) before making their own intertemporal choice again (Self3, light green solid line). For this participant, their preference shifted towards that of Patient Other ('Patient_{DKL}'). The positive_{DKL} value here means that the participant's preference became more similar to that of the Patient Other after observing their preference.

R3.7: p. 15/27 - It would be helpful to provide a better description of what the emotional sensitivity subscale measures – what do higher scores on the measure mean, what are sample items on the scale, etc. Also, why did you not focus on the social subscale? Wouldn't this make more sense, given that you are examining social influence? Also, switching to discuss "emotional motivation" in the manuscript seems to conflate these two concepts.

Response: Thank you for offering us an opportunity to further elucidate the emotional sensitivity subscale of the Apathy-Motivation Index (AMI) and elaborate on theoretical accounts of the relationship between emotional motivation and susceptibility to social influence. The emotional sensitivity subscale of the AMI measures people's tendency to experience both positive and negative affection. Each item is reverse scored, meaning that higher scores on this subscale represent lower emotional motivation (i.e., greater emotional apathy). The scale comprises items such as 'I feel sad or upset when I hear bad news', 'I feel awful if I say something insensitive', and 'I feel bad when I hear an acquaintance has an accident or illness'. On the other hand, the social motivation subscale of the AMI captures people's tendency to engage in social interactions, and every item is also reverse scored, meaning that higher scores on this subscale indicate lower social motivation. It includes items like 'I suggest activities for me and my friends to do', 'I start conversations with random people', and 'I enjoy doing things with people I have just met'.

From the sample items, one may discern that the emotional sensitivity subscale measures how likely people's emotional experiences are triggered by events (often within social contexts), while the social motivation subscale measures how likely people initiate a social interaction. In our study, participants were asked to passively observe other people's decisions and to learn about their preferences, a scenario more aligned with the constructs examined in the emotional sensitivity subscale. In addition, emotional motivation has been linked to affective empathy, the capacity to resonate with the emotions of others, as suggested by the theoretical framework of motivated empathy and empirical work (Contreras-Huerta et al., 2020; Lockwood et al., 2017; Zaki, 2014). A key aspect of affective empathy involves self-other control, which entails maintaining a balance between representing the self and the other (de Guzman et al., 2016). Flexibly representing the self and the other is essential in the dynamics of social influence (Garvert et al., 2015; Nicolle et al., 2012). Therefore, we hypothesised that individual variations in emotional motivation and affective empathy might explain individual differences in susceptibility to social influence. Our new factor analysis also substantiated this conjecture, showing that the scores from the factor 'Affective empathy & emotional motivation' were significantly correlated with susceptibility to impulsive social influence in older adults (see the **Response** to R3.5).

R3.8: Perhaps I missed it, but are the similarity ratings (described on page 27) discussed anywhere?

Responses: The results of the similarity rating have been reported on pages 12-13, lines 240-242, in the original manuscript (we apologise that the layout may have made it hard to notice):

Results:

... This finding aligns with the observation that participants reported feeling more similar to patient others compared to impulsive ones ($b = 1.20$, 95% CI = [0.53 1.78], $Z = 3.62$, $P < 0.001$).

Decision letter and referee reports: second round

Dear Mr Su,

Your manuscript titled "Older adults are relatively more susceptible to impulsive social influence than young adults" has now been seen by our reviewers, whose comments appear below. In light of their advice I am delighted to say that we are happy, in principle, to publish a suitably revised version in *Communications Psychology*.

We therefore invite you to revise your paper one last time to address the remaining concerns of our reviewers and a list of editorial requests. The limitations section can serve to mention remaining criticisms from the reviewers. At the same time we ask that you edit your manuscript to comply with our format requirements and to maximise the accessibility and therefore the impact of your work.

EDITORIAL REQUESTS:

SUBMISSION INFORMATION:

OPEN ACCESS:

* **DATA AVAILABILITY:**

(link Redacted)

Best regards,

Jennifer Bellingtier

Jennifer Bellingtier, PhD
Senior Editor
Communications Psychology

REVIEWERS' EXPERTISE:

Reviewer #1 development/decision-making

Reviewer #2 social decision making/computational models

REVIEWERS' COMMENTS:

Reviewer #1 (Remarks to the Author):

Thank you for the opportunity to re-review this manuscript. The authors have been very responsive to my (and other reviewers') comments, and I only have one remaining concern. The current title, "Older adults are relatively more susceptible to impulsive social influence than young adults," still implies that older adults are the age group deviating from the norm when, in fact, it is the younger adults who are doing so. A simple flip, "Younger adults are relatively less susceptible to impulsive social influence than older adults," emphasizes that younger adults are the group who are deviating.

Reviewer #2 (Remarks to the Author):

I appreciate the time and effort that the authors have dedicated to this comprehensive revision. The manuscript is much more comprehensible now and the additional analysis strengthen the results. The PPC that I recommended was conducted very carefully and the findings support the modeling results.

What I did not find satisfactorily addressed the conceptual issue of involvement of Theory of Mind. The authors argue that they did not manipulate ToM levels and that ToM does not explain the differences effects of learning about other's preferences observed in younger and older adults. But that was not my argument in the first place. The task itself does not feature direct social influence (despite the claim in previous publications): at no point during the task, the participant is shown the actual choices of another person (which happens to be not real after all). Rather, the participant makes choices *for* another agent and learns about the agents delay preferences only through feedback. It really doesn't get more indirect than this! Rather it seems pretty obvious to me that in this task the participant learns to construct a mental model of the other agent's delay preference and this self-evident fact puts this task in the ToM domain (which btw do not always have to feature different levels). Of course, the finding that self choices are modulated by the mental model learned for the other agent can be labeled as "social influence", but this is really different from what the field normally understands as social influence, which features observing the behavior of others and subsequently adapting one's own behavior accordingly. Here, it's all the "head" of the participant, no social behavior (i.e. choices of others) are ever observed.

I am not saying that the findings of this paper are uninteresting and not worth publishing. But I do think that re-framing the task in terms of ToM would open up a new set of fascinating questions of how we interact with our mental models, and these questions remain unexpressed and unanswered here.

Another important aspect that was not properly addressed in the revision was the question about the cognitive mechanisms that are involved in modulating self choices after learning about others, and this shortcoming persists regardless of the whether the task is framed as "social influence" or "mentalizing": through model-free and model-informed analyses the authors demonstrate the shift in own delay preference, but they do not provide any explanation of how, let alone why, the modulation is realized psychologically and computationally.

Author Responses: second round.

Older adults are relatively more susceptible to impulsive social influence than young adults

Reviewer #1 (Remarks to the Author):

Thank you for the opportunity to re-review this manuscript. The authors have been very responsive to my (and other reviewers') comments, and I only have one remaining concern. The current title, "Older adults are relatively more susceptible to impulsive social influence than young adults," still implies that older adults are the age group deviating from the norm when, in fact, it is the younger adults who are doing so. A simple flip, "Younger adults are relatively less susceptible to impulsive social influence than older adults," emphasizes that younger adults are the group who are deviating.

Response: Thank you for taking the time to re-review our work and for providing us with your helpful comments. In the study, we do not assume whether being influenced by others is normative or not, especially considering there might be different norms for impulsive versus patient influence. We believe the current title effectively reflects this perspective and is agnostic to what the 'normative' amount of social influence is and therefore opted to retain the revised title, which now describes that the comparison is relative to younger adults as requested in R1.

Reviewer #2 (Remarks to the Author):

I appreciate the time and effort that the authors have dedicated to this comprehensive revision. The manuscript is much more comprehensible now and the additional analysis strengthen the results. The PPC that I recommended was conducted very carefully and the findings support the modeling results.

What I did not find satisfactorily addressed the conceptual issue of involvement of Theory of Mind. The authors argue that they did not manipulate ToM levels and that ToM does not explain the differences effects of learning about other's preferences observed in younger and older adults. But that was not my argument in the first place. The task itself does not feature direct social influence (despite the claim in previous publications): at no point during the task, the participant is shown the actual choices of another person (which happens to be not real after all). Rather, the participant makes choices *for* another agent and learns about the agents delay preferences only through feedback. It really doesn't get more indirect than this! Rather it seems pretty obvious to me that in this task the participant learns to construct a mental model of the other agent's delay preference and this self-evident fact puts this task in the ToM domain (which btw do not always have to feature different levels). Of course, the finding that self choices are modulated by the mental model learned for the other agent can be labeled as "social influence", but this is really different from what the field normally understands as social influence, which features observing the behavior of others and subsequently adapting one's own behavior accordingly. Here, it's all the "head" of the participant, no social behavior (i.e. choices of others) are ever observed.

I am not saying that the findings of this paper are uninteresting and not worth publishing. But I do think that re-framing the task in terms of ToM would open up a new set of fascinating questions of how we interact with our mental models, and these questions remain unexpressed and unanswered here.

Another important aspect that was not properly addressed in the revision was the question about the cognitive mechanisms that are involved in modulating self choices after learning about others, and this shortcoming persists regardless of the whether the task is framed as "social influence" or "mentalizing": through model-free and model-informed analyses the authors demonstrate the shift in own delay preference, but they do not provide any explanation of how, let alone why, the modulation is realized psychologically and computationally.

Response: We really appreciate your re-review of our work. We agree that placing this experimental paradigm within the framework of theory of mind could offer valuable insights into why people's preferences are swayed by observing or inferring others' behaviours. However, we want to emphasise that, regardless of whether people construct mental models of others, their behaviours are still influenced by the behaviours of others, whether observed or implied. Additionally, we acknowledge that the computational models used to quantify social influence may not fully explain how and why people shift their preferences and behaviours, which represents a limitation of our study and highlights the need for further investigation. We have added these observations to our discussion and limitations section:

Discussion:

Future studies could also examine how much insight older adults have into their greater influence by impulsive others to understand whether and how such effects can be modified. There is also increasing empirical interest in the relationship between metacognition and mentalising, with possible computational and neural overlap between them⁹⁵⁻⁹⁷. In our task, another possible interpretation is that being influenced by others' beliefs is related to one's theory of mind ability, given that the choices of others were inferred rather than directly observed. However, we found people displayed different susceptibility to impulsive and patient social influence. Even if theory of mind is indeed involved, it should theoretically be engaged in both scenarios. In addition, studies of older adults often suggest reduced or preserved theory of mind⁹⁸, whereas we found relatively enhanced susceptibility to social influence in older adults. Future studies could dissect and dissociate how and why metacognition, mentalising, and social influence drive any differences between older and young adults. Situating our experimental paradigm within the framework of metacognition or theory of mind might offer invaluable insights into how and why people's preferences are swayed by observing or inferring the behaviours of others.

Limitations:

In addition to these novel findings, there are also limitations. Firstly, we focussed on a specific type of social influence related to economic preferences. Future research could examine a broader range of social influences that may differ between young and older adults. Secondly, our study was cross-sectional by design. Longitudinal studies are needed incorporating mid-life samples to understand how susceptibility to social influence evolves throughout adulthood and account for possible non-linear changes. Third, we used validated computational models of social influence. However, these models do not incorporate social-cognitive aspects such as theory of mind that may account for why people shift their preferences and behaviours and individual differences between people. Finally, the abstract nature of the experimental design may have missed some of the complexities inherent in real-world social influence scenarios⁷⁴. Future research could consider examining social in everyday experiences.